# Tick hemocytes have a pleiotropic role in microbial infection and arthropod fitness

Agustin Rolandelli [1], Hanna J. Laukaitis-Yousey [1], Haikel N. Bogale[2,7], Nisha Singh[1,8], Sourabh Samaddar [1], Anya J. O'Neal[1,9], Camila R. Ferraz [1], Matthew Butnaru[3,4], Enzo Mameli [3,5], Baolong Xia[3], M. Tays Mendes [1], L. Rainer Butler [1,10], Liron Marnin[1], Francy E. Cabrera Paz[1], Luisa M. Valencia[1], Vipin S. Rana[6], Ciaran Skerry[1], Utpal Pal [6], Stephanie E. Mohr [3], Norbert Perrimon [3,4], David Serre [1,2] & Joao H. F. Pedra [1] ✉

Uncovering the complexity of systems in non-model organisms is critical for understanding arthropod immunology. Prior efforts have mostly focused on Dipteran insects, which only account for a subset of existing arthropod species in nature. Here we use and develop advanced techniques to describe immune cells (hemocytes) from the clinically relevant tick *Ixodes scapularis* at a single-cell resolution. We observe molecular alterations in hemocytes upon feeding and infection with either the Lyme disease spirochete *Borrelia burgdorferi* or the rickettsial agent *Anaplasma phagocytophilum*. We reveal hemocyte clusters exhibiting defined signatures related to immunity, metabolism, and proliferation. Depletion of phagocytic hemocytes affects *hemocytin* and *astakine* levels, two *I. scapularis* hemocyte markers, impacting blood-feeding, molting behavior, and bacterial acquisition. Mechanistically, astakine alters hemocyte proliferation, whereas hemocytin affects the c-Jun N-terminal kinase (JNK) signaling pathway in *I. scapularis*. Altogether, we discover a role for tick hemocytes in immunophysiology and provide a valuable resource for comparative biology in arthropods.

The evolution of arthropod immune systems is shaped by the environment, longevity, nutrition, microbial exposure, development, and reproduction[1,2]. Yet, our current understanding of immunology is centered around Dipteran insects, which reflects an incomplete depiction of arthropod biological networks existing in nature[3–9]. Divergent signaling modules prevail between Dipterans and other arthropods, highlighting the complex and dynamic features occurring across evolutionarily distant species. One example is the immune deficiency (IMD) pathway, a network analogous to the tumor necrosis factor receptor (TNFR) in mammals[10]. Some components of the canonical IMD pathway are not observed in arachnids, although this immune pathway remains functional and responsive to microbial infection[3–7]. Another instance is the Janus Kinase/Signal Transducer and Activator of Transcription (JAK/STAT) signaling pathway, which in flies is elicited by unpaired cytokine-like molecules[11], whereas in ticks host-derived interferon (IFN)-γ facilitates Dome1-mediated

[1]Department of Microbiology and Immunology, University of Maryland School of Medicine, Baltimore, MD, USA. [2]Institute for Genome Sciences, University of Maryland School of Medicine, Baltimore, MD, USA. [3]Department of Genetics, Blavatnik Institute, Harvard Medical School, Boston, MA, USA. [4]Howard Hughes Medical Institute, Chevy Chase, MD, USA. [5]Department of Microbiology, National Emerging Infectious Diseases Laboratories, Boston University School of Medicine, Boston, MA, USA. [6]Department of Veterinary Medicine, University of Maryland, College Park, MD, USA. [7]Present address: Rancho BioSciences, San Diego, CA, USA. [8]Present address: Department of Biotechnology, School of Energy Technology, Pandit Deendayal Energy University; Knowledge Corridor, Gandhinagar, Gujarat, India. [9]Present address: Immunology Program, Memorial Sloan Kettering Cancer Center, New York, NY, USA. [10]Present address: Department of Genetics, Blavatnik Institute, Harvard Medical School, Boston, MA, USA. ✉e-mail: jpedra@som.umaryland.edu

activation[8,9]. This biochemical network enhances tick blood meal acquisition and development while inducing the expression of antimicrobial components[8,9]. Thus, investigating the plasticity of immune networks in evolutionarily divergent organisms may reveal discrete aspects of arthropod biology.

Hard ticks are ancient arthropods that serve as major vectors of human and animal pathogens, including the Lyme disease spirochete *Borrelia burgdorferi* and the rickettsial agent *Anaplasma phagocytophilum*[12]. Selective pressure caused by longevity, exclusive hematophagy in all life stages, long-term dispersal (adaptation to multiple environments) and exposure to an array of microbes make ticks distinct from other hematophagous arthropods[13,14]. While it is acknowledged that the immune system plays an important role in vector competence, our understanding of the mechanisms by which immune resistance balances microbial infection in ticks remains fragmented[5–9].

Hemocytes are specialized arthropod immune cells that function in both cellular and humoral responses. These cells circulate through the hemolymph and are in contact with tissues within the arthropod body cavity[15]. Historically, tick hemocytes have been categorized according to their cellular morphology and ultrastructural features[16–18]. Although useful, this classification is incomplete because the ontogeny, plasticity, and molecular features of hemocytes during hematophagy and infection remain ill-defined. Here, we used bulk and single-cell RNA sequencing (scRNA-seq) coupled with Clustered Regularly Interspaced Short Palindromic Repeats (CRISPR) activation, RNA interference (RNAi), RNA-fluorescence in situ hybridization (RNA-FISH), and immune cell depletion through treatment with clodronate liposomes to reveal distinct features of hemocyte clusters in the blacklegged tick. We identify marker genes for each hemocyte cluster, predict lineages, and unveil specific biological signatures related to immunity, proliferation, and metabolism during hematophagy. We further characterize an immune cluster that expresses *hemocytin* (*hmc*) and *astakine* (*astk*). Manipulation of phagocytic hemocytes and the expression of *hmc* or *astk* affects bacterial acquisition, feeding and molting. Overall, we highlight the exquisite biology of ticks, demonstrate the canonical and non-canonical roles of immune cells in an obligate hematophagous ectoparasite, and provide a critical resource for comparative biology of arthropods.

## Results

### Blood-feeding induces molecular signatures in *I. scapularis* hemocytes related to immunity, metabolism, and proliferation

Ticks rely solely on blood as a source of essential metabolites, ingesting ~100 times their body mass per meal[19]. During feeding, extensive modifications and tissue rearrangements are necessary to accommodate and digest such large volumes of blood[19]. Considering that hemocytes are the circulating cells in the hemolymph, we hypothesized that these immune cells sense and respond to physiological and microbial exposure during tick infestation on vertebrate hosts. We optimized a protocol to collect hemocytes from *I. scapularis* nymphs, a clinically relevant stage in the blacklegged tick, and identified three common hemocyte morphotypes reported in the literature (Fig. 1a, Supplementary Fig. 1)[16–18]. Prohemocytes, considered the stem cell-like hemocyte population, were the smallest cells, with a round or oval shape between 5.7 and 12.5 μm (average diameter of 8.5 μm). The cytoplasm was minute (high nuclear/cytoplasmic ratio) and homogeneous, with no apparent protrusions or

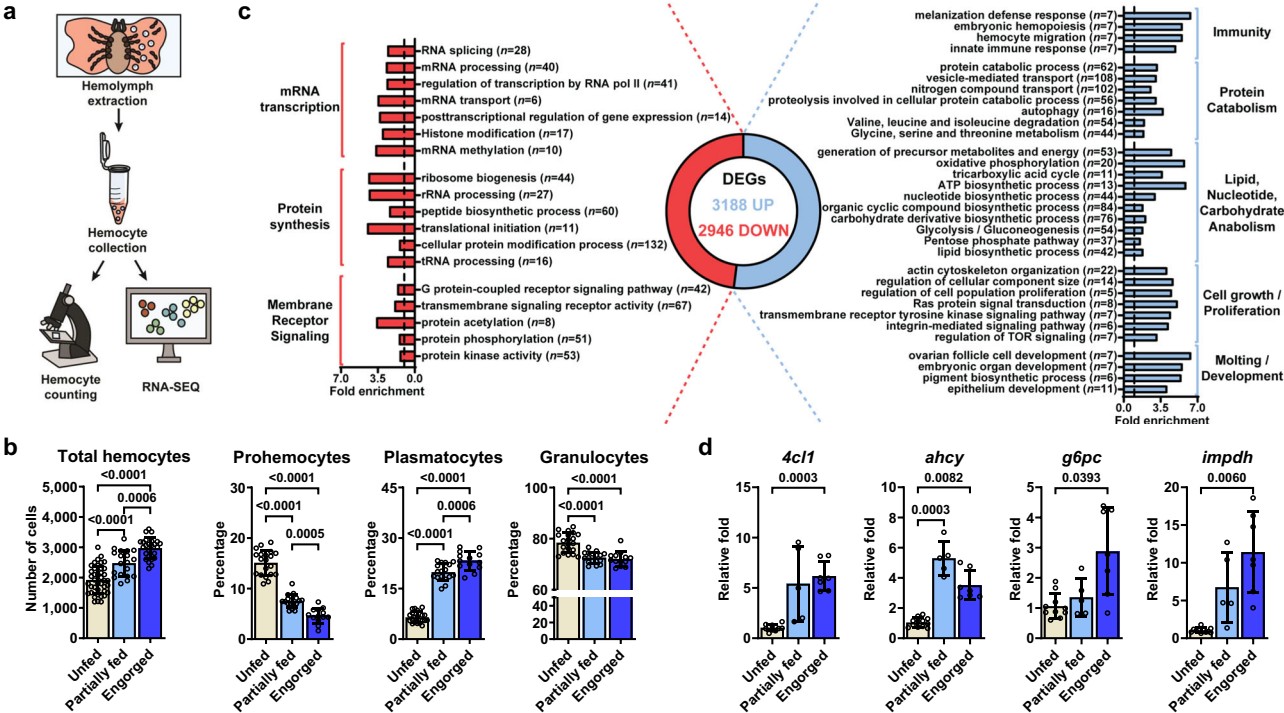

**Fig. 1 | Blood-feeding induces alterations in *I. scapularis* hemocytes. a** Schematic representation of the hemocyte-enriched collection procedure. **b** Total number of hemocytes (*n* = 40, 25 and 19) and percentages of different morphotypes (prohemocytes, plasmatocytes and granulocytes; *n* = 20, 13 and 17 for all cases) from unfed (ivory), partially fed (light blue) or engorged (dark blue) nymphs. **c** Functional enrichment analysis of the differentially expressed genes (DEGs) in hemocyte-enriched samples from engorged ticks (blue; Up) compared to unfed ticks (red; Down). Fold enrichment of significant categories (FDR < 0.05) is depicted. The number of DEGs per category is shown in parentheses. **d** The expression of key genes upregulated during feeding in hemocyte-enriched samples from unfed (ivory), partially fed (light blue) and engorged (dark blue) ticks was evaluated by RT-qPCR (*n* = 11, 5 and 7 for all cases; with 40–80 pooled ticks per sample).

**b, d** Results are represented as mean ± SD. At least three independent experiments were performed. Statistical significance was evaluated by Brown-Forsythe ANOVA test, and significant *p*-values (< 0.05) are displayed in the figure. Source data are provided as a Source Data file. *4cl1* = 4-coumarate-CoA ligase 1; *ahcy* = adenosylhomocysteinase B; *g6pc* = glucose-6-phosphatase 2; *impdh* = inosine-5'-monophosphate dehydrogenase 1.

granules (Supplementary Fig. 1a, d). Granulocytes were round or oval shaped cells with diameters ranging from 10.4 to 22.8 μm (average of 16.7 μm). The position of the nucleus varied, appearing most often near the periphery of the cell. Their cytoplasm was filled with dark-blue or violet stained granules or vacuoles (Supplementary Fig. 1b, d). Plasmatocytes varied in size (ranged from 14.8 to 31.2 μm, with an average of 21.3 μm) and shape (oval, ameboid-like, pyriform), and had cytoplasmic protrusions or pseudopodia-like structures. The cytoplasm was clear and had few dark-blue or violet stained granules or vacuoles. The nucleus was located either near the center or the periphery of the cell (Supplementary Fig. 1c, d). We then investigated the impact of blood-feeding on hemocytes originating from *I. scapularis* nymphs. We observed an increased quantity of total hemocytes upon mammalian feeding (Fig. 1b). The percentage of plasmatocytes increased in engorged ticks, whereas we noticed a decline in the proportion of prohemocytes and granulocytes in repleted nymphs compared to unfed (Fig. 1b). Altogether, we demonstrate the impact of hematophagy on the distribution of tick hemocyte morphotypes.

Next, we aimed to examine global transcriptional changes induced by hematophagy through bulk RNA-seq. We collected hemocyte-enriched samples from unfed and engorged nymphs and observed drastic changes in gene expression, with a total of 6134 differentially expressed genes (DEGs) (Fig. 1c; Supplementary Data 1). Samples from unfed *I. scapularis* nymphs were enriched for housekeeping genes, including those involved in mRNA transcription (e.g., RNA splicing, mRNA processing, histone modification), protein synthesis (*e.g.*, peptide biosynthetic process, cellular protein modification process, ribosome biogenesis) and membrane receptor signaling pathways (*e.g.*, transmembrane signaling receptor activity, G protein-coupled receptor signaling pathway, protein kinase activity) (Fig. 1c; Supplementary Data 2). In contrast, hemocyte-enriched samples from engorged ticks exhibited an overrepresentation of gene signatures related to immunity, metabolic pathways, cell proliferation/growth, and arthropod molting/development (Fig. 1c; Supplementary Fig. 2; Supplementary Data 2). Key genes modulated during feeding, including *4-coumarate-CoA ligase 1* (*4cl1*), *adenosylhomocysteinase B* (*ahcy*), *glucose-6-phosphatase 2* (*g6pc*), *inosine-5′-monophosphate dehydrogenase 1* (*impdh*), *Brahma chromatin remodeling complex subunit osa* (*osa*), *runt-related transcription factor 1* (*runx*) and *frizzled-5* (*frizzled*), were independently validated using RT-qPCR (Fig. 1d; Supplementary Fig. 3). Notably, the overall expression levels of these genes in hemocyte-enriched samples from partially fed ticks were intermediate compared to those in unfed and engorged ticks, suggesting a transitional phenotype. Collectively, this dataset indicated that *I. scapularis* hemocytes exhibit a dynamic genetic program during hematophagy.

## Defining *I. scapularis* hemocyte clusters by scRNA-seq

To uncover whether the transcriptional changes observed through bulk RNA-seq are accompanied with heterogeneity among hemocytes, we performed scRNA-seq. We collected hemocyte-enriched samples from (1) unfed nymphs, (2) engorged nymphs fed on uninfected mice, and engorged nymphs fed on mice infected with either (3) the rickettsial pathogen *A. phagocytophilum* or (4) the Lyme disease spirochete *B. burgdorferi*. After stringent quality controls, we profiled a total of 20,432 cells (unfed = 4630; engorged uninfected = 6000; engorged *A. phagocytophilum*-infected = 6287; engorged *B. burgdorferi*-infected = 3515), with a median of 744 unique molecular identifiers (UMIs), 261 genes and 6.2% of mitochondrial transcripts per cell across conditions (Supplementary Fig. 4). Consistent with the bulk RNA-seq results, the principal component analysis (PCA) identified distinct distributions between unfed and fed conditions, reinforcing the notion that significant cellular and/or transcriptional changes occur following a blood meal (Supplementary Fig. 5).

Following unsupervised clustering, we identified seven clusters in unfed ticks and thirten clusters in engorged nymphs (Supplementary Fig. 6). Based on similarities in marker gene profiles, two clusters in the unfed and two clusters in the engorged datasets were merged (Supplementary Data 3, 4). Thus, six and twelve clusters remained, respectively, with each cluster expressing a unique set of cell type-defining genes (Fig. 2a-c, Supplementary Data 5-7). One cluster in each dataset showed high expression of gut-associated genes *cathepsinB*, *cathepsinD* and *boophilinH2* (Fig. 2a-c, Supplementary Data 5-7) and two additional clusters in the engorged dataset had gene expression profiles indicative of cuticle and salivary glands tissues (Fig. 2a-c, Supplementary Data 5 and 7). Therefore, these clusters were excluded from subsequent analysis. We assigned putative functions to the top 50 DEGs per cluster using information for tick genes in VectorBase, the sequence homology to *D. melanogaster* in FlyBase and the presence of functionally annotated protein domains in InterPro (Fig. 2c-e, Supplementary Data 6, 7). We also performed functional enrichment analysis based on gene ontology (GO) using the entire list of DEGs for each cluster (Supplementary Data 8, 9). Altogether, we defined five hemocyte clusters from unfed and nine clusters in engorged *I. scapularis* nymphs.

## scRNA-seq uncovers hemocyte adaptations to hematophagy in *I. scapularis*

The molecular features and differentiation process of *I. scapularis* hemocytes is currently unknown. Thus, based on GO enrichment and marker gene profiles, we were able to characterize clusters of hemocytes shared by both unfed and engorged ticks (Supplementary Figs. 7, 8, Supplementary Data 5-7). The Immune 1 cluster showed high expression of genes related to phagocytosis or cytoskeleton organization; coagulation and agglutination functions, such as lectins (*hemocytin*, *techylectin-5A*), chitin-binding and clotting proteins; and secreted proteins related to immunity, such as *astakine*, *microplusin*, mucins and cystatin domain peptides (Fig. 2c-e, Supplementary Data 5-7). The Immune 2 cluster displayed genes encoding secreted proteins involved in immunity, such as antimicrobial peptides (AMPs) and clotting related peptides (Fig. 2c-e, Supplementary Data 5-7). The Proliferative 1 cluster was enriched with mitochondrial genes, characteristic of stem cells in ancient arthropods, such as crayfish[20]. This cell cluster also had high expression of genes related to actin polymerization, cell proliferation and differentiation (Fig. 2c-e, Supplementary Data 5-7). The Proliferative 2 cluster displayed a high percentage of transcription factors, RNA binding proteins and genes related to actin dynamics. Marker genes for this cluster included several genes involved in hormone-related responses, suggesting they may be responsive to ecdysteroids synthesized after a blood meal (Fig. 2c-e, Supplementary Data 5-7). Lastly, both datasets had a Transitional cluster indicative of intermediate subtypes (Fig. 2c-e, Supplementary Data 5-7).

Four hemocyte clusters were only observed in engorged *I. scapularis* ticks. The Immune 3 cluster displayed an enrichment in secreted proteins and genes related to immune functions. This cluster was enriched for chitinases, matrix and zinc metalloproteinases, peptidases, and actin binding proteins, which have roles related to wound healing or tissue rearrangement. Several glycine-rich proteins (GRPs), commonly associated with antimicrobial properties or structural proteins, were also present (Fig. 2c-e, Supplementary Data 5, 7). The Immune 4 cluster showed an overrepresentation of genes related to protein degradation, immune function, and cell proliferation (Fig. 2c-e, Supplementary Data 5, 7). Thus, we posit that the Immune 4 cluster represents an intermediate state between the Immune 2 and the Proliferative 2 clusters. Two clusters displayed an enrichment for genes related to metabolic functions. The Metabolism 1 cluster represented genes involved in sulfonation of proteins, lipids and glycosaminoglycans, transmembrane solute transporter, nucleotide, and

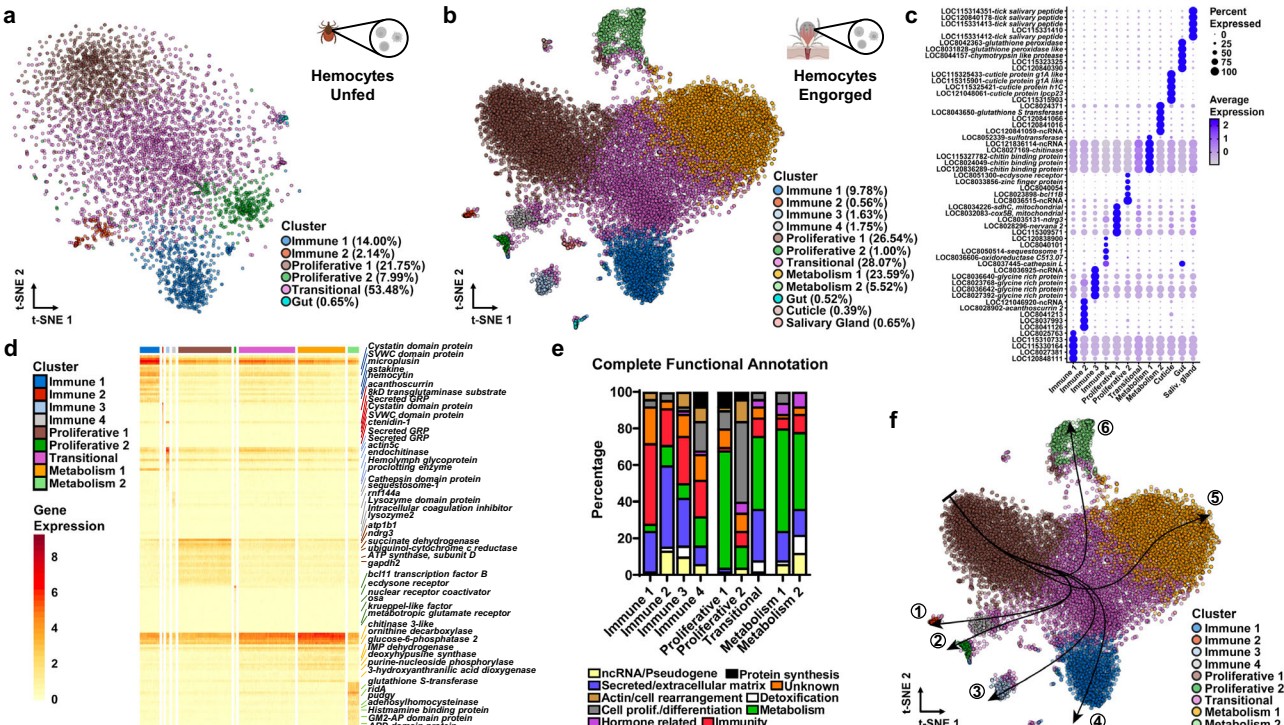

**Fig. 2 | scRNA-seq uncovers hemocytes with immune, proliferative and metabolic signatures in *I. scapularis*.** Hemocyte-enriched samples pooled from individual unfed (n = 90) or engorged uninfected, *A. phagocytophilum*- or *B. burgdorferi*-infected *I. scapularis* ticks (n = 50 for each) were collected immediately post-detachment from the host. t-Distributed Stochastic Neighbor Embedding (t-SNE) plot clustering of samples collected from (**a**) unfed (4,630 cells) and (**b**) engorged (15,802 cells) nymphs. The engorged *t*-SNE includes cells from uninfected (6000 cells), *A. phagocytophilum*-infected (6287 cells) and *B. burgdorferi*-infected (3515 cells) ticks. **c** Dot plot of the top 5 marker genes present in clusters from engorged ticks based on average gene expression. Color intensity demarks gene expression level, while the size of the dot indicates the percentage of cells within individual clusters expressing the corresponding gene. **d** Heatmap depicting the

expression of marker genes for hemocyte subtypes from engorged ticks. The top 20 marker genes per cluster based on gene expression are included, with representative genes highlighted. **e** The top 50 marker genes from each hemocyte cluster were manually annotated using the VectorBase, FlyBase, and UniProt publicly available databases. The percentage of predicted functional categories, such as ncRNA/pseudogenes (yellow), protein synthesis (black), secreted/extracellular matrix (blue), unknown (orange), actin/cell rearrangement (brown), detoxification (white), cell proliferation/differentiation (grey), metabolism (green), hormone-related (purple), and immunity (red) are shown. **f** Pseudotime analysis using slingshot defined six hemocyte lineages (indicated by arrows) in engorged ticks. Tick images in (**a**, **b**) were created with BioRender.com.

---

protein metabolism (Fig. 2c–e, Supplementary Data 5, 7). The Metabolism 2 cluster displayed genes related to detoxification, histamine binding, lipid metabolism, methionine and juvenile hormone metabolism (Fig. 2c–e, Supplementary Data 5, 7).

Based on these findings, we predicted hemocyte ontogeny using pseudotime analysis, which orders hemocyte clusters based on their gene expression profiles, enabling us to infer developmental lineages (Fig. 2f)[21]. We found six trajectories considering the cluster Proliferative 1 as a stem cell-like subpopulation. Lineage 1 and 2 ended with the Immune 2 and Proliferative 2 clusters, respectively, with the Immune 4 cluster serving as an intermediate state. Lineages 3 and 4 gave rise to the Immune 3 and Immune 1 clusters, respectively. Finally, two lineage trajectories ended with the metabolic clusters, Metabolism 1 and Metabolism 2. Overall, our findings suggest the presence of an oligopotent subpopulation that differentiates into more specialized subtypes involved in immune and metabolic functions, a process evoked by hematophagy.

### Bacterial infection of *I. scapularis* ticks alters the molecular profile of hemocytes

The impact of bacterial infection on subtypes of tick hemocytes remains elusive. Thus, we collected hemocytes from *I. scapularis* nymphs fed on uninfected, *A. phagocytophilum*- or *B. burgdorferi*-infected mice and determined morphotype percentages. During infection with the rickettsial agent *A. phagocytophilum*, a relative decrease in prohemocytes and increase in plasmatocytes was noted

(Fig. 3a). Conversely, only a slight decrease in the proportion of pro-hemocytes was observed during infection with the Lyme disease spirochete *B. burgdorferi* (Fig. 3a). No difference in total hemocyte numbers was observed across infection conditions (Supplementary Fig. 9). However, partitioning the engorged scRNA-seq datasets by treatments revealed a reduction in the Transitional cluster with an expansion in the Metabolism 1 cluster during *B. burgdorferi* infection (Fig. 3b, c). We next analyzed transcriptional changes at the cellular level in engorged uninfected nymphs compared to engorged infected ticks. Hemocyte clusters were grouped according to three molecular programs: "Immune" (Immune 1-4), "Proliferative" (Proliferative 1-2 and Transitional) and "Metabolism" (Metabolism 1-2). During *A. phagocytophilum* infection, we identified 177 DEGs within the Proliferative clusters, 53 DEGs in the Metabolism clusters, and 5 DEGs among the Immune clusters (Fig. 3d, Supplementary Data 10). Conversely, during *B. burgdorferi* infection, we detected 244 DEGs within the Proliferative clusters, 81 DEGs among the Metabolism clusters, and 12 DEGs associated with the Immune clusters (Fig. 3d, Supplementary Data 10). In total, we identified 188 and 257 unique DEGs across all hemocyte clusters during *A. phagocytophilum* and *B. burgdorferi* infection, respectively. Notably, 11 genes were differentially expressed during both bacterial infections and shared amongst all cluster groupings, in which 36.4% (4 out of 11) were marker genes of the Immune 1 cluster (Fig. 3e). Collectively, our results defined specific hemocyte subpopulations and genes that are differentially expressed in *I. scapularis* in response to *A. phagocytophilum* or *B. burgdorferi* infection.

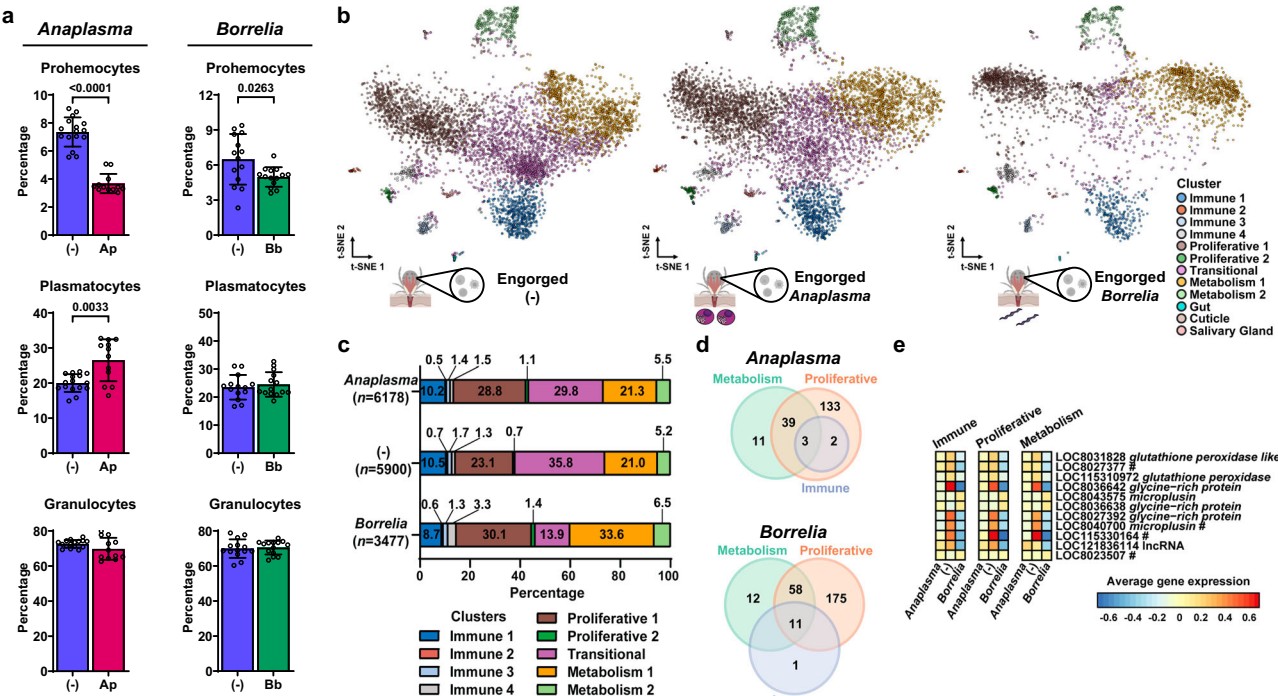

**Fig. 3 | Bacterial infection alters hemocyte subtypes and their gene expression.**
**a** Hemocyte morphotypes (prohemocytes, plasmatocytes and granulocytes) in *I. scapularis* nymphs fed on *A. phagocytophilum-* (Ap, pink) or *B. burgdorferi-* (Bb, green) infected mice compared to uninfected [(-), dark blue] (*n* = 16 and 12; *n* = 14 and 14, respectively). Results are presented as mean ± SD. A minimum of two independent experiments were conducted. Statistical significance was determined using an unpaired two-tailed *t* test with Welch's correction, and significant *p*-values (< 0.05) are displayed in the figure. **b** t-Distributed Stochastic Neighbor Embedding (t-SNE) plot clustering of cells collected from the hemolymph of uninfected (6000 cells), *A. phagocytophilum-* (6287 cells) or *B. burgdorferi*-infected (3515 cells) *I. scapularis* nymphs. **c** Percentage of hemocyte clusters in uninfected, *A. phagocytophilum-* or *B. burgdorferi*-infected ticks. **d** Venn diagram illustrating the number of differentially expressed genes (DEGs) between groups of hemocyte clusters during *A. phagocytophilum* (top) or *B. burgdorferi* (bottom) infection compared to uninfected ticks. Hemocyte clusters were categorized into three molecular programs: "Immune" (Immune 1-4), "Proliferative" (Proliferative 1-2 and Transitional) and "Metabolism" (Metabolism 1-2). DEGs were determined using pairwise comparisons against uninfected. **e** Heatmap representing the average expression patterns of DEGs altered during infection and shared between all 3 cluster groups. For each DEG, the mean average across all experimental conditions was centered at zero for each hemocyte group. The # symbol indicates Immune 1 marker genes. Source data are provided as a Source Data file. Tick images in (b) were created with BioRender.com. (-) = Uninfected. *Anaplasma* = *A. phagocytophilum*. *Borrelia* = *B. burgdorferi*.

## Phagocytic hemocytes contribute to *A. phagocytophilum* infection and tick fitness

The preceding findings suggested that the Immune 1 hemocyte cluster represented a subpopulation of cells that responded to bacterial infections in ticks (Fig. 3d). Therefore, our focus shifted to two marker genes from the Immune 1 hemocyte cluster: *hemocytin* and *astakine* (Fig. 4a, Supplementary Data 6, 7). *Hemocytin* is homologous to *hemolectin* in *D. melanogaster* and von Willebrand factors of mammals[22,23]. *Hemocytin* encodes for a large multidomain adhesive protein involved in clotting, microbial agglutination and hemocyte aggregation[22–25]. Conversely, astakine is a cytokine-like molecule present in chelicerates and crustaceans and is homologous to vertebrate prokineticins[26–29]. Astakine also induces hemocyte proliferation and differentiation of immune cells[26–29]. *Hemocytin* and *astakine* were broadly expressed in the Immune 1 cluster of *I. scapularis* hemocytes (Fig. 4a). Validating our scRNA-seq results, we confirmed the expression of *hemocytin* and *astakine* in hemocyte-enriched samples obtained from unfed ticks using RNA-FISH, illustrating that these genes serve as markers for the Immune 1 cluster (Fig. 4b, Supplementary Fig. 10). Furthermore, we noted an upregulation of *hemocytin* and *astakine* in hemocyte-enriched samples collected from engorged ticks, a pattern also observed in other markers of the Immune 1 cluster (Fig. 4c, Supplementary Data 11). These findings suggested that blood-feeding expanded this hemocyte subtype and/or upregulated the expression of its marker genes in *I. scapularis* ticks.

*Hemolectin* is a known marker of phagocytic plasmatocytes in *Drosophila*[30–32], and phagocytic hemocytes expressing *hemocytin* are present in crustaceans[20,33,34]. Therefore, we explored the phagocytic potential of the Immune 1 cluster by employing clodronate liposomes (CLD), which have been used to deplete phagocytic hemocytes in flies, mosquitoes, and, more recently, ticks[35–38]. We found that CLD treatment led to a 36% reduction in the total number of hemocytes in *I. scapularis* nymphs (Fig. 4d). We observed an increase in the proportion of granulocytes along with a decrease in the percentages of prohemocytes and plasmatocytes compared to ticks treated with empty liposomes (Fig. 4e). Interestingly, the expression of both Immune 1 marker genes, *hemocytin* and *astakine*, also decreased after CLD treatment, implying the Immune 1 cluster is phagocytic (Fig. 4f). Taken together, CLD treatment effectively reduced the number of phagocytic hemocytes in *I. scapularis*, particularly impacting prohemocytes, plasmatocytes and cells associated with the Immune 1 cluster.

Phagocytosis serves as a pivotal immune mechanism against invading microbes. Previous studies have demonstrated that the depletion of phagocytic immune cells can influence the survival of arthropods following infection with either Gram-positive or Gram-negative bacteria[35–38]. However, the impact of depleting phagocytic hemocytes on the acquisition of tick-borne bacteria remains unclear. Thus, we further investigated the effects of CLD treatment during *A. phagocytophilum* infection, as this intracellular microbe is known to interact with *I. scapularis* hemocytes shortly after uptake[39]. Our findings revealed a reduction in *A. phagocytophilum* load in engorged ticks after injection with CLD compared to controls (Fig. 4g), with no differences in tick attachment (Supplementary Fig. 11), suggesting that

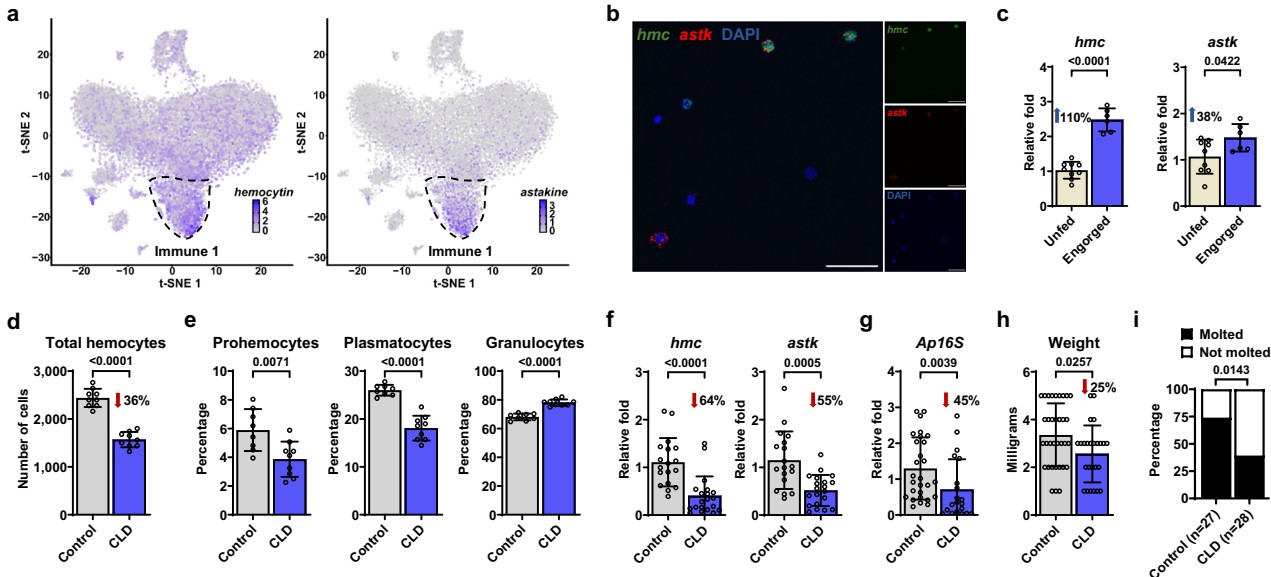

**Fig. 4 | Phagocytic hemocytes contribute to _A. phagocytophilum_ infection and fitness in ticks. a** Expression patterns of _hemocytin_ (left) and _astakine_ (right) on t-Distributed Stochastic Neighbor Embedding (t-SNE) plots of hemocyte-enriched samples from engorged nymphs. Their highest expression is denoted in the Immune 1 cluster (outlined). **b** RNA FISH image of _I. scapularis_ hemocytes labeled for _hemocytin_ (_hmc_, green), _astakine_ (_astk_, red), and nuclei (DAPI). White scale bars indicate a length of 50 μm. Refer to Supplementary Fig. 10 for control images. **c** RT-qPCR evaluation of _hemocytin_ (_hmc_; $n = 9$ and 6) and _astakine_ (_astk_; $n = 9$ and 6) expression in hemocyte-enriched samples collected from unfed (ivory) or engorged (dark blue) ticks (with 40–80 pooled ticks per sample). **d-i** Ticks were subjected to microinjection with clodronate (CLD) or empty liposomes (Control) and subsequently fed on either (**d-f, h–i**) uninfected or (**g**) _A. phagocytophilum_-infected mice. **d** Total hemocyte counts ($n = 9$ and 9) and (**e**) morphotype percentages (prohemocytes, plasmatocytes and granulocytes; $n = 8$ and 9 in all cases) were assessed in the hemolymph of individual ticks. **f** RT-qPCR analysis of _hemocytin_ (_hmc_; $n = 18$ and 20) and _astakine_ (_astk_; $n = 18$ and 20) expression, and (**g**) _A. phagocytophilum_ load in individual ticks ($n = 28$ and 20). Bacterial quantification was based on the expression of _A. phagocytophilum_ 16 s rRNA (_Ap16S_) gene. **h** Weight measurements of engorged nymphs ($n = 34$ and 25). **i** Percentage of nymphs that molted to adults. Results are represented as (**c-h**) mean ± SD or as (**i**) a percentage from the total. A minimum of two independent experiments were conducted. Statistical significance was evaluated by (**c-f**) an unpaired two-tailed _t_ test with Welch's correction, (**g-h**) two-tailed Mann–Whitney U test or (**i**) by a Fisher exact test, and significant _p_ values (< 0.05) are displayed in the figure. Source data are provided as a Source Data file.

phagocytic hemocytes promote either the acquisition or proliferation of _A. phagocytophilum_ during blood-feeding.

Reports on Dipteran model organisms have demonstrated that hemocytes play roles beyond immunity, including tissue communication, clearing apoptotic cells during molting and development, and serving as vehicles for molecules[40–44]. However, whether hemocytes have non-immune roles in ticks remains unknown. Our sequencing results revealed an enrichment in functions related to metabolism, cell proliferation, and development after a blood meal, suggesting that hemocytes may participate in the feeding or molting processes in ticks (Figs. 1, 2). To investigate this further, we recorded these physiological processes after injecting ticks with CLD. Although no differences in attachment were observed (Supplementary Fig. 11), engorged ticks treated with CLD weighed significantly less compared to the control treatment, indicating that phagocytic hemocytes are required for proper hematophagy (Fig. 4h). Furthermore, the number of nymphs that successfully molted to adults was significantly lower in CLD-treated ticks (Fig. 4i). Collectively, these results indicated that phagocytic hemocytes pleiotropically impacted tick immunity, feeding, and molting in _I. scapularis_. Importantly, the observed correlation between changes in physiological parameters and the reduced expression of the Immune 1 markers, _hemocytin_ and _astakine_, during CLD treatment suggested a potential regulatory role for these genes in various aspects of tick physiology, prompting us to explore this hypothesis further.

### Astakine induces hemocyte proliferation and differentiation in _I. scapularis_

We found that hematophagy induces hemocyte proliferation and differentiation (Fig. 1b) while upregulating _astakine_ expression (Fig. 4c).

Thus, we investigated whether _astakine_ was directly implicated in the proliferation or differentiation of _I. scapularis_ hemocytes. We microinjected increasing amounts of recombinant astakine (rAstk) in unfed nymphs and observed a dose-dependent increase in the total number of hemocytes (Fig. 5a). Specifically, we measured a decrease in the percentage of prohemocytes and an increase in plasmatocytes (Fig. 5b). These findings matched alterations observed during normal blood-feeding (Fig. 1b). Interestingly, we also detected an increase in tick IDE12 cell numbers in vitro following treatment with rAstk, supporting a role of astakine in inducing cell proliferation in hemocyte-like cells (Supplementary Fig. 12). To corroborate these results, we then silenced _astakine_ by microinjecting unfed nymphs with small-interfering RNA (siRNA) before feeding on uninfected mice. We utilized siRNA to knockdown gene expression, as genome editing through CRISPR has only recently been introduced for adult germline manipulation and possesses low efficiency in ticks[45]. We recovered 41% less hemocytes and observed an increase in the percentage of prohemocytes with a decrease in plasmatocytes from engorged ticks when _astakine_ was silenced (Fig. 5c–e). Therefore, we determined that astakine acts on hematopoietic processes in the ectoparasite _I. scapularis_.

Our previous results showed alterations in the percentage of hemocytes by the CLD treatment, which influenced bacterial infection and tick physiology. Given that astakine also alters hemocyte composition in ticks, we then investigated whether decreasing the expression of _astakine_ affects _A. phagocytophilum_ infection, hematophagy or ecdysis in _I. scapularis_. Accordingly, we observed a significant reduction in weight, molting success, and _A. phagocytophilum_ burden in ticks silenced for _astakine_ compared to the control treatment, without differences in tick attachment

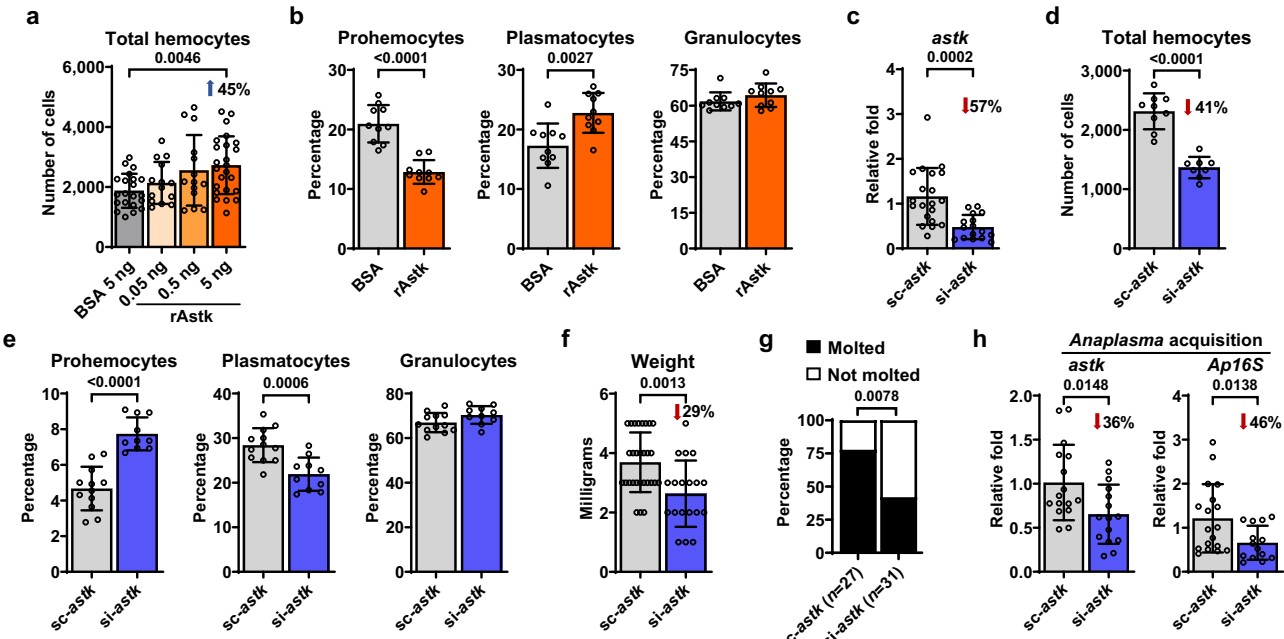

**Fig. 5 | Astakine (astk) promotes hemocyte proliferation and differentiation in I. scapularis. a** Total hemocyte counts in the hemolymph of unfed *I. scapularis* nymphs subjected to microinjection with increasing amounts of rAstk (orange) or BSA (grey) as a control (*n* = 21, 14, 14 and 24). **b** Percentage of hemocyte morphotypes (prohemocytes, plasmatocytes and granulocytes; *n* = 10 and 10 in all cases) in the hemolymph of unfed nymphs following microinjection with 5 ng rAstk (orange) compared to BSA controls (grey). **c**–**h** Ticks were subjected to microinjection with *astk* siRNA (si-*astk*; blue) or scrambled RNA (sc-*astk*; grey) and subsequently fed on either (**c**–**g**) uninfected or (**h**) *A. phagocytophilum*-infected mice. **c** Efficacy of *astk* silencing (*n* = 20 and 16), (**d**) total number of hemocytes (*n* = 9 and 8) and (**e**) percentage of hemocyte morphotypes in individual ticks (*n* = 12 and 10 in all cases).

**f** Weight measurements of engorged nymphs (*n* = 29 and 19). **g** Percentage of nymphs that molted to adults. **h** RT-qPCR assessment of *astk* silencing efficiency (*n* = 16 and 15) and *A. phagocytophilum* load (*n* = 18 and 14) in individual infected ticks. Bacterial quantification was based on the expression of *A. phagocytophilum* 16 s rRNA (*Ap16S*) gene. Results are represented as (**a**–**f**, **h**) mean ± SD or as (**g**) a percentage from the total. A minimum of two independent experiments were conducted. Statistical significance was evaluated by (**a**) one-way ANOVA with Dunnett's multiple comparisons test; (**b**–**f**, **h**) an unpaired two-tailed *t* test with Welch's correction or (**g**) by a Fisher exact test, and significant *p*-values (< 0.05) are displayed in the figure. Source data are provided as a Source Data file. rAstk = recombinant astakine; BSA = bovine serum albumin.

(Fig. 5f–h; Supplementary Fig. 13). Corroborating these findings, *A. phagocytophilum* load was also lower in *astakine*-silenced tick cells in vitro (Supplementary Fig. 14). Overall, we demonstrated that astakine regulates hemocyte composition in *I. scapularis*, which affects not only bacterial infection but also feeding and ecdysis, supporting a pleiotropic role of hemocytes in ticks.

## Hemocytin affects the JNK pathway of *I. scapularis*

The broad expression of *hemocytin* detected in *I. scapularis* hemocytes (Fig. 4a), and its homology to a *Drosophila* gene used as a marker of phagocytic immune cells, prompted us to explore whether *hemocytin* could be involved in hemocyte proliferation or differentiation, phagocytosis or immune signaling in ticks. Surprisingly, no differences were observed in hemocyte numbers or subtype proportions between ticks injected with siRNA targeting *hemocytin*, nor in the phagocytic capacity of *hemocytin*-silenced tick cells (Supplementary Figs. 15, 16). We then asked whether *hemocytin* could act as an immune pathway regulator in *I. scapularis*. We focused on the IMD and the c-Jun N-terminal kinase (JNK) pathways, as previous reports indicated the importance of these molecular networks during bacterial infection of ticks[5,6].

After transfection with siRNA targeting *hemocytin*, we found a decrease in JNK phosphorylation in *hemocytin*-silenced tick cells, without alteration in Relish cleavage (Fig. 6a, b). To complement our findings, we overexpressed *hemocytin* in tick cells through CRISPR activation (CRISPRa). CRISPRa has been widely used to enhance the expression of an endogenous locus, employing a catalytically inactive Cas9 (dCas9) fused with transcription activators and single guide RNAs (sgRNAs) that direct the modified enzyme to the promoter region of a gene of interest[46]. However, so far none of the CRISPRa effectors have

been tested in tick cell lines and no endogenous promoter has been identified for genetic expression of sgRNAs. We optimized reagents and developed a protocol to up-regulate *hemocytin* in ISE6 cells using two rounds of nucleofection with different expression plasmids. The first plasmid expressed dCas9-VPR paired with neomycin resistance under the control of the CMV promoter. The second plasmid expressed either a sgRNA specific to the promoter region of *hemocytin* (*hmc*-sgRNA) or a control sgRNA (ctrl-sgRNA) driven by an endogenous RNA polymerase III promoter (Fig. 6c). Tick ISE6 cells were used as a platform for CRISPRa given the lower expression of *hemocytin* compared to IDE12 cells (Supplementary Fig. 17). Strikingly, we detected a 277% increase in the expression of *hemocytin* in dCas9⁺ cells transfected with the *hmc*-sgRNA compared to dCas9⁺ cells transfected with the ctrl-sgRNA (Fig. 6d). Importantly, we also noticed elevated levels of both *jun* (the transcription factor for the JNK pathway) and *jnk* expression (Fig. 6d).

To corroborate our results in vivo, we microinjected unfed ticks with a scrambled control or siRNA targeting *hemocytin* and allowed them to feed on naïve mice. Upon repletion, we measured the expression of *jun* and *relish*. Consistently, we found that reduction in *hemocytin* expression led to a decrease in *jun* levels in ticks, without affecting *relish* expression (Fig. 6e), uncovering a role for hemocytin in the activation of the JNK pathway in *I. scapularis*.

Finally, building on our previous findings demonstrating that blood-feeding and infection alter the expression of *hemocytin* in ticks (Fig. 4c, Supplementary Data 9), coupled with the documented roles of hemocytin as an agglutinating factor and the JNK pathway in enhancing organismal growth and metabolism in insects[47], we delved deeper into the relationship between *hemocytin* expression and physiological parameters in *I. scapularis*. We detected a decrease in weight and

molting success to adulthood in *hemocytin*-silenced ticks that fed on uninfected mice compared to scrambled controls, with no differences in tick attachment (Fig. 6f, g; Supplementary Fig. 18). Additionally, we noted that silencing *hemocytin* increased *A. phagocytophilum* load both in vivo and in vitro (Fig. 6h; Supplementary Fig. 19), supporting an antimicrobial role for *hemocytin*. Collectively, our findings indicated that tick hemocyte subtypes and their associated marker genes play a critical role in *I. scapularis* immunophysiology.

## Discussion

The immune response against microbial infection has been extensively studied in Dipteran insects, but these species do not fully account for the variation in fundamental processes that exists in arthropods[3-9,14]. In this study, we expand on the previous morphological classification of tick hemocytes. We characterized hemocytes at a single-cell resolution in combination with orthogonal approaches to uncover hemocyte subtypes and molecular markers in *I. scapularis*. Our findings revealed immune and metabolic alterations of tick hemocytes in response to blood-feeding and microbial infection. In contrast to mosquitoes[48,49], *I. scapularis* hematophagy prompts the emergence of new hemocyte clusters and distinct biological signatures.

Hemocytes with immune-related roles exhibited over-representation of genes involved in peptide secretion, agglutination, clotting, extracellular matrix remodeling and protein degradation. These data suggest that these clusters not only participate in antimicrobial responses, but also aid in the extensive internal remodeling of tissues required during blood meal acquisition[19]. Given the established immunological function of plasmatocytes and granulocytes in

ticks, we speculate that the immune clusters represent these morphotypes[16-18]. Metabolic hemocytes were enriched in solute transport, lipid and protein metabolism, histamine binding and detoxification genes. These molecular features posit an involvement in the feeding process by metabolizing nutrients and xenobiotics present in the blood meal. We also identified hemocytes specialized in cellular proliferation, including cells expressing genes involved in ecdysteroid biosynthesis, essential for regulating molting and development[6,50]. We considered these proliferative clusters, prohemocytes, which have been categorized as a stem cell-like hemocyte subpopulation[16-18].

With the advent of cutting-edge technologies, hemocyte subtypes have been characterized at the molecular level in various arthropods, facilitating a broad comparative analysis of the evolution of the immune system[20,30-34,48,49,51-53]. In the model organism *D. melanogaster*, three hemocyte subtypes exist: (1) plasmatocytes, responsible for phagocytosis, clotting, and AMP secretion; (2) crystal cells, involved in the melanization process by synthesizing pro-phenoloxidase (proPO); and (3) lamellocytes, which encapsulate foreign bodies[30-32]. Interestingly, ticks do not seem to trigger melanization involving the proPO cascade, as proPO genes have not been found in the *I. scapularis* genome[54]. Furthermore, markers typically associated with specific hemocyte subtypes in *D. melanogaster* were not detected in *I. scapularis* hemocyte clusters, except for *hemocytin*. These discrepancies in marker genes are also observed in closely related species. For instance, there is a lack of *hemocytin* expression or lamellocyte markers in hemocytes from *Anopheles gambiae* compared to *D. melanogaster*. Additionally, there is the presence of *ProPO* genes in several hemocyte subtypes of mosquitoes as opposed to the specific pattern in crystal

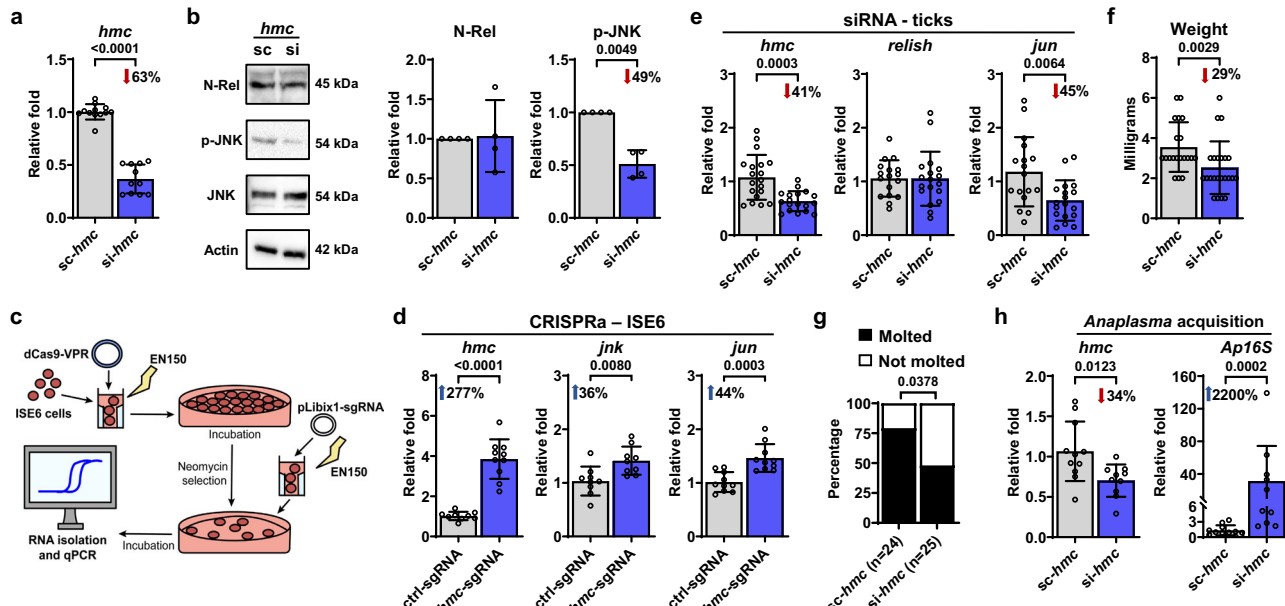

**Fig. 6 | *Hemocytin (hmc)* positively impacts the JNK pathway in *I. scapularis*.**
**a, b** Tick cells were transfected with either *hmc* siRNA (si-*hmc*) or scrambled RNA (sc-*hmc*). **a** Efficiency of *hmc* silencing in IDE12 cells (*n* = 12 and 11). **b** Representative western blot (left) of N-Rel and p-JNK during treatment with sc-*hmc* (lane 1) or si-*hmc* (lane 2). N-Rel and p-JNK protein expression was quantified (right) in si-*hmc* (blue) or sc-*hmc* (grey) IDE12 cells (*n* = 4). Data were normalized to the scrambled control, with N-Rel values relative to Actin and p-JNK values to JNK. A representative blot from four experiments is shown. Uncropped blots containing the molecular weight markers are supplied in the Source data. **c** Overview of CRISPRa-mediated *hmc* overexpression in ISE6 cells. **d** RT-qPCR analysis of *hmc* (left; *n* = 9 and 10), *jnk* (middle; *n* = 9 and 9) and *jun* (right; *n* = 10 and 10) expression in dCas9⁺ ISE6 cells transfected with either a single guide RNA (sgRNA) specific to the promoter region of *hemocytin* (*hmc*-sgRNA, blue) or a random sgRNA (ctrl-sgRNA, grey). **e–h** Ticks were subjected to microinjection with *hmc* siRNA (si-*hmc*; blue) or scrambled siRNA

(sc-*hmc*; grey) and subsequently fed on either (**e–g**) uninfected or (**h**) *A. phagocytophilum*-infected mice. **e** RT-qPCR analysis of *hmc* (left; *n* = 19 and 18), *relish* (middle; *n* = 17 and 18) and *jun* (right; *n* = 17 and 18) expression in engorged ticks. **f** Weight measurements of engorged nymphs (*n* = 20 and 23). **g** Percentage of nymphs that molted to adults. **h** RT-qPCR assessment of *hmc* silencing efficiency (*n* = 11 and 10) and *A. phagocytophilum* load (*n* = 11 and 10) in individual infected ticks. Bacterial quantification was based on the expression of *A. phagocytophilum* 16 s rRNA (*Ap16S*) gene. Results are represented as mean ± SD. A minimum of two independent experiments were performed. Statistical significance was evaluated by (**a, b, d, e**) an unpaired two-tailed t-test with Welch's correction; (**f, h**) a two-tailed Mann–Whitney U test or (**g**) by a Fisher exact test, and significant *p*-values (< 0.05) are displayed in the figure. Source data are provided as a Source Data file. N-Rel = cleaved Relish; p-JNK = phosphorylated JNK; JNK = c-Jun N-terminal kinase.

cells of flies[30–32,48,49]. Collectively, genetic divergences appear to be common occurrences in hemocytes during evolution, likely arising from differences in longevity, metabolic resources and exposure to microbes.

Despite the absence of universally conserved genetic markers, hemocytes in arthropods exhibit shared subtypes with functions related to immunity, proliferation or metabolism[20,30–34,48,49,51,53]. For instance, specialized immune clusters of hemocytes involved in combating infection have been identified, such as "megacytes" in *A. gambiae*[55]. Furthermore, the specialization of hemocyte subtypes in activities related to proliferation and metabolism is evident in both Hexapoda and Crustacea[20,30–32,34,51]. Accordingly, we and others have noted significant changes in the numbers of tick hemocyte and subtype distributions following a blood meal[35]. This is also the case for mosquitoes, in which blood-feeding increases granulocyte numbers, likely by triggering granulocyte proliferation and prohemocyte differentiation[49,56,57]. Therefore, while hemocytes are recognized as specialized immune cells in arthropods, it is evident that they have acquired distinct roles through evolution that extend beyond immunity.

For *I. scapularis*, the presence of *A. phagocytophilum* or *B. burgdorferi* does not cause adverse effects on tick viability, suggesting that co-evolutionary mechanisms have developed[1,2]. This was further exemplified by the lack of new hemocyte clusters emerging during infection, highlighting an opportunity for comparative biology in arthropods. However, we established specific molecular programs in hemocytes altered during bacterial infection of *I. scapularis*. Avoidance of strong cellular responses upon microbial infection may conceivably be related to a discrete evolutionary balance between immunity and survival in hematophagous arthropods. Using CLD to deplete phagocytic hemocytes, we observed a reduction in the percentage of prohemocytes and plasmatocytes, which correlated with a decrease in *A. phagocytophilum* load. Consistent with previous evidence showing that blocking phagocytosis in hemocytes reduces *A. phagocytophilum* dissemination from the midgut to other tick tissues[39], our results affirm the essential role of plasmatocytes in the infection process by this intracellular bacterium. Supporting our findings, it was recently reported that *Ehrlichia chaffeensis* relies on phagocytic hemocytes for systemic infection in the Lone Star tick, *Amblyomma americanum*[38]. Hence, tick-borne intracellular bacteria have evolved mechanisms to exploit hemocytes for dissemination, unveiling an essential process for successful vector transmission.

CLD treatment resulted in a decrease of *hemocytin* and *astakine* levels, two marker genes of the Immune 1 cluster. Mechanistically, we uncovered a function for *astakine* in tick hematopoiesis, similar to observations in crustaceans[26]. Consistent with the CLD experiments, silencing *astakine* led to reduced percentages of plasmatocytes and decreased *A. phagocytophilum* load in *I. scapularis*, emphasizing this morphotype as a key participant during bacterial infection. Future studies determining the cellular source, receptors, and signaling pathways of astakine will reveal mechanisms driving proliferation and differentiation of immune cells in ticks. Conversely, silencing of *hemocytin* in ticks led to an elevated *A. phagocytophilum* burden during hematophagy independent of phagocytosis. Supporting these findings, knockdown of *hemocytin* in the hard tick *A. americanum* led to an increase in the intracellular bacterium *E. chaffeensis*[53]. Interestingly, *hemocytin* was shown to be involved in microbial agglutination in insects[23–25]. Whether *hemocytin* acts directly as an agglutination factor in ticks is yet to be determined. Furthermore, both *hemocytin* and *astakine* were decreased in hemocytes from the tick *Amblyomma maculatum* during infection with the intracellular bacterium *Rickettsia parkeri*, suggesting that these genes may contribute to microbial infection in other tick species[35].

Hemocytes are important for organogenesis, development, clearing of apoptotic cells and tissue communication in insects[40–44]. We showed that changes in phagocytic hemocytes, as well as alterations in *astakine* and *hemocytin* levels, impacted the fitness of *I. scapularis*. Acari exclusively utilize ecdysteroids, synthesized from cholesterol, for molting[6,50]. Considering the transcriptional profile induced by feeding and the circulatory nature of hemocytes, we hypothesize that these cells are involved in the internal reorganization of tissues needed during hematophagy and ecdysis. Our findings indicated the involvement of hemocytes in ecdysteroid synthesis and transport, illustrated by alterations in lipid metabolism-related genes after a blood meal. Specifically, our dataset highlights an essential role of plasmatocytes in tick feeding and molting. This assertion is supported by a decrease in plasmatocytes upon *astakine* knockdown or CLD treatment, which led to a diminished tick weight and a reduced number of nymphs transitioning to adulthood. Conversely, *hemocytin* knockdown disrupted these processes without altering hemocyte numbers or their phagocytic capacity. Intriguingly, we demonstrated a role of *hemocytin* in modulating the JNK pathway in ticks. Given its conservation across arthropods and its recognized function as a pro-survival mechanism in *Drosophila*[47], we postulate that the JNK pathway plays a key role in tick organismal growth. Future investigations should focus on identifying the molecular connections between *hemocytin* and JNK signaling.

In summary, we leveraged the power of systems biology in combination with reductionist approaches to shed light on the complexity and dynamic attributes of biological processes associated with immune cells of an obligate hematophagous ectoparasite. Our findings uncovered canonical and non-canonical roles of tick hemocytes, which might have evolved because of evolutionary associations of *I. scapularis* with its vertebrate hosts and microbial exposure. The hemocyte dataset presented in this work, along with the identified cell type-specific marker genes and experimental tools, will facilitate future mechanistic studies in ticks. A few examples may include: (1) the use of marker genes to identify specific hemocyte subsets susceptible to infection or capable of microbial phagocytosis; (2) CRISPRa to study a gene of interest in ticks through overexpression; and (3) the development of antibodies against specific cell clusters for sorting hemocyte subtypes during distinct physiological conditions. Collectively, our open-access resource will foster hypothesis-driven research that could be adapted to construct an integrated atlas of the medically relevant tick *I. scapularis*.

## Methods

### Ethics statement

Mouse breeding, weaning and tick experiments were performed under guidelines from the NIH (Office of Laboratory Animal Welfare (OLAW) assurance numbers A3200-01) and pre-approved by the Institutional Biosafety (IBC-00002247) and Animal Care and Use (IACUC-0119012) committee of the University of Maryland School of Medicine.

### Cell lines

The *I. scapularis* IDE12 and ISE6 cell lines were obtained from Dr. Ulrike Munderloh at University of Minnesota. Cells were cultured in L15C300 medium supplemented with 10% heat inactivated fetal bovine serum (FBS, Sigma), 10% tryptose phosphate broth (TPB, Difco), 0.1% bovine cholesterol lipoprotein concentrate (LPPC, MP Biomedicals) at 34 °C in a non-$CO_2$ incubator. ISE6 cells were grown to confluency in capped T25 flasks (Greiner bio-one) and either seeded at $3 \times 10^6$ cells/well in 6-well plates (Millipore Sigma) or $3 \times 10^5$ cells/well in 24-well plates (Corning). IDE12 cells were also grown to confluency in T25 flasks and either seeded at $1 \times 10^6$ cells/well in 6-well plates (Millipore Sigma) or $3 \times 10^5$ cells/well in 24-well plates (Corning). The human leukemia cell line, HL-60, was obtained from ATCC and maintained in RPMI-1640 medium with L-Glutamine (Quality Biological) supplemented with 10% FBS (Gemini Bio-Products) and 1% GlutaMax (Gibco), in vented T25 flasks (CytoOne) at 37 °C and 5% $CO_2$. All cell cultures were confirmed to be *Mycoplasma*-free (Southern Biotech).

## Bacteria, ticks, and mice

*A. phagocytophilum* strain HZ was grown as previously described in HL-60 cells at 37 °C, using RPMI medium supplemented with 10% FBS and 1% Glutamax[58]. Bacterial numbers were calculated using the following formula: number of infected HL-60 cells × 5 morulae/cell × 19 bacteria/morulae × 0.5 recovery rate[59]. To determine the percentage of infection, cells were spun onto microscope slides using a Epredia™ Cytospin™ 4 Cytocentrifuge (Thermo Scientific), stained using the Richard-Allan Scientific™ three-step staining (Thermo Scientific) and visualized by light microscopy using a Primo Star™ HAL/LED Microscope (Carl Zeiss). For in vitro experiments, bacteria were purified in L15C300 media by passing infected cells through a 27-gauge bent needle and centrifugation, as previously described[58]. Isolated bacteria were inoculated onto tick cells at a multiplicity of infection (MOI) of 50 for 48 h. For in vivo experiments, a total of $1 \times 10^7$ *A. phagocytophilum*-infected cells/mL was resuspended in 100 μL of 1X phosphate-buffered saline (PBS) and intraperitonially injected into mice. Infection progressed for 6 days prior to tick placement.

*B. burgdorferi* B31 clone MSK5 (passage 3–5) was cultured in Barbour-Stoenner Kelly (BSK)-II medium supplemented with 6% normal rabbit serum (Pel-Freez) at 37 °C. Before mouse infection, plasmid profiling was performed as described elsewhere[60]. To determine the infectious inoculum, spirochetes were enumerated under a 40X objective lens using a dark-field microscope (Carl Zeiss™ Primo Star™ Microscope) by multiplying the number of bacteria detected per field x dilution factor x $10^{6}$[58,60]. For intradermal inoculation, spirochetes were washed, resuspended in 50% normal rabbit serum at a concentration of $1 \times 10^6$ *B. burgdorferi*/mL and anesthetized C3H/HeJ mice were injected with 100 μL of inoculum ($1 \times 10^5$ total *B. burgdorferi*). Infected mice were maintained for at least 14 days prior to tick placement.

*I. scapularis* nymphs were obtained from Oklahoma State University and University of Minnesota. Upon arrival, ticks were maintained in a Percival I-30BLL incubator at 23 °C with 85% relative humidity and a 12/10 h light/dark photoperiod regimen. Age matched (6-10 weeks), male C57BL6 and C3H/HeJ mice were supplied by Jackson Laboratories or the University of Maryland Veterinary Resources. For all tick experiments, capsules were attached to the back of mice using a warmed adhesive solution made from 3 parts gum rosin (Millipore-Sigma) and 1 part beeswax (Thermo Scientific) 24 h prior to tick placement. All experiments were done using C57BL6 mice, except for the ones including *B. burgdorferi* infection (C3H/HeJ genetic background).

## Hemocyte collection

Hemolymph was collected from *I. scapularis* nymphs in L15C300 + 50% FBS on ice from individual unfed, partially fed, or engorged ticks placed on uninfected, *A. phagocytophilum*- or *B. burgdorferi*-infected mice, using non-stick RNase-free 1.5 mL microtubes (Ambion) and siliconized pipet tips. Briefly, ticks were immobilized on a glass slide using double-sided tape (3 M) and covered in a sphere of 10 μL of L15C300 + 50% FBS. Forelegs were incised at the tarsus, and forceps were used to gently apply pressure to the tick body to release hemolymph into the media. Enumeration of total hemocytes was determined for each tick using a Neubauer chamber. For differentiation of hemocyte morphotypes, 100 μL of L15C300 was added in a microtube for each tick before preparation on Fisherbrand™ Superfrost™ Plus microscope slides (Fisher Scientific) using a Epredia™ Cytospin™ 4 Cytocentrifuge (Thermo Scientific). Afterwards, hemocyte-enriched samples were stained using the Richard-Allan Scientific™ three-step staining (Thermo Scientific) and evaluated morphologically under a Primo Star™ HAL/LED Microscope (Carl Zeiss). Images were acquired using an Axiocam 305 color camera (Carl Zeiss) with the ZEN software (Carl Zeiss), which was used for measuring cell diameters.

For RNA extraction, hemocyte-enriched samples were collected as mentioned above using PBS (4 μL for unfed and 6 μL for engorged ticks) instead of media. For each biological replicate, hemocytes from 80 unfed or 40 engorged ticks (retrieved immediately post-detachment from the host) were pooled in 700 μL of TRIzol.

## Bulk RNA sequencing

Three independent hemocyte-enriched sample collections were performed for each condition (unfed and engorged), as mentioned above. After collection, RNA was extracted from TRIzol following the manufacturer's instruction with the addition of 1 μg of glycogen per sample. The RNA integrity (RIN) was assessed for each sample using a Eukaryote Total RNA Nano Chip Assay on a Bioanalyzer (Agilent). RIN values ranged between 7.6 and 9.4. Strand-specific, dual unique indexed libraries for sequencing on Illumina platforms were generated using the NEBNext® Ultra™ II Directional RNA Library Prep Kit for Illumina® (New England Biolabs). The manufacturer's protocol was modified by diluting adapters 1:30 and using 3 μL of this dilution. The size selection of the library was performed with SPRI-select beads (Beckman Coulter Genomics). Glycosylase digestion of the adapter and $2^{nd}$ strand was performed in the same reaction as the final amplification. The sample input was polyA-enriched. Libraries were assessed for concentration and fragment size using the DNA High Sensitivity Assay on the LabChip GX Touch (Perkin Elmer). The resulting libraries were sequenced on a multiplexed paired end 100 bp Illumina NovaSeq 6000 S4 flowcell, generating an average of 92 million read pairs per sample. RIN analysis, library preparation and sequencing were completed by Maryland Genomics at the Institute for Genome Sciences, University of Maryland School of Medicine. One sample was excluded from subsequent analysis due to considerably smaller library size compared to other biological replicates.

After sequencing, all reads were aligned to the *I. scapularis* genome (GCF_016920785.2_ASM1692078v2) using HISAT v2.0.4[14,61]. Potential PCR duplicates were removed by keeping only one of the read pairs mapped to the same genomic location of the same strand and then tallied the number of "unique reads" mapped to each annotated *I. scapularis* gene. The generated count matrix was used as input for differential expression analysis with edgeR v3.36.0[62]. During data normalization, only transcripts with at least 10 counts per million (cpm) in two or more samples were kept for downstream analysis. Differentially expressed genes (DEGs) were defined based on the following thresholds: (1) a false discovery rate (FDR) < 0.05, and (2) $-0.5146 < \logFC > 0.3785$, corresponding to at least a 30% reduction or increase in gene expression. All sequencing reads were deposited into the NCBI Sequence Read Archive under the BioProject accession PRJNA906572.

## Single-cell RNA sequencing

For single-cell RNA sequencing (scRNA-seq), hemocyte-enriched samples were collected from individual unfed, engorged, or *A. phagocytophilum*- or *B. burgdorferi*-infected ticks and each condition pooled in 300 μL of L15C300 + 50% FBS (n = 90 for unfed and n = 50 for engorged ticks retrieved immediately post-detachment from the host). Cells were filtered through a 40 μm cell strainer (Millipore Sigma) and enriched for live cells using OptiPrep density gradient centrifugation[30]. Briefly, 200–300 μL of the filtered solution was overlaid onto 2 mL of OptiPrep™ Density Gradient Medium (1.09 g/mL in 1X PBS; Sigma) and centrifugated at 1000 x g for 10 min at 4 °C, using a swinging bucket without breaking. After centrifugation, 200 μL of the interphase was collected in a new microtube and spun at 1000 x g for 10 min at 4 °C. Cells were concentrated in 30 μL of 1X PBS + 0.2 U RNasin® Plus Ribonuclease Inhibitor (Promega), from which 5 μL were used for measuring viability and cell numbers by trypan blue exclusion assay. The final concentration of cells ranged from 350 to 500 cells/μL, with viability ranging between 95.7% and 98.5%.

An estimated 5000–7000 cells per condition were loaded on a 10X Genomics Chromium controller and four individually barcoded 3'-end scRNA-seq libraries were prepared according to the

manufacturer's protocol. Each library was then sequenced on an Illumina high-output sequencer to generate a total of 816,376,790 75-bp paired-end reads. To process all scRNA-seq reads, a custom pipeline previously used to analyze single-cell data from *Plasmodium* parasites similar to 10X Genomics Cell Ranger software was adapted[63]. First, only reads longer than 40 bp after trimming any sequences downstream of 3' polyadenylation tails were kept. These reads were then mapped to the *I. scapularis* genome (GCF_016920785.2_ASM1692078v2) using HISAT version 2.0.4[14,61]. Out of the mapped reads, "unique reads" were identified by keeping only one of the reads that had identical 16-mer 10X Genomics barcode, 12-mer unique molecular identifier, and mapped to the same genomic location of the same strand. Finally, the 10X Genomics barcodes were used to separate reads derived from each individual cell and count the number of unique reads mapped to each annotated *I. scapularis* gene (from transcription start site to 500 bp after the annotated 3'-end). All sequencing reads are deposited into the NCBI Sequence Read Archive under the BioProject accession PRJNA905678 and the code for mapping is available through the GitHub website (https://github.com/Haikelnb/scRNASeq_Analysis).

The scRNA-seq libraries were combined into a single dataset using scran v1.26.2[64]. Genes containing "40s_ribosomal" and "60s_ribosomal" were removed from the entire dataset. Based on Principal Component Analysis (PCA), unfed and engorged hemocyte-enriched samples were analyzed independently for downstream analyses. Low quality cells were removed with the following thresholds: less than 600 unique reads, less than 150 mapped genes, and 30% or higher mitochondrial transcripts. The remaining transcriptomes were normalized by first calculating size factors via scran functions quickCluster and computeSumFactors and computing normalized counts for each cell with logNormCounts function in scater v1.22.0[65]. For downstream analysis, highly variable genes were selected using getTopHVGs before performing PCA and t-distributed Stochastic Neighbor Embedding (t-SNE) projections. Clustering was conducted using a kmeans value of 20. Differential gene expression between clusters was calculated using the find marker function in scran v1.26.2[64]. The R package slingshot v2.2.1 was used to perform pseudotime inference where trajectories began from the "Proliferative 1" cluster, based on predicted function[21]. To determine differences in gene expression between each infected condition compared to the reference factor (uninfected), clusters were grouped based on annotated function (*e.g.*, metabolism, proliferation, immune) and MAST v1.24.1 was used to test for significance under the Hurdle model adjusting for the cellular detection rate[66]. The code for the analysis is available through the GitHub website (https://github.com/HLaukaitisJ/PedraLab_hemocyte_scRNAseq).

### Functional annotation and enrichment analysis

Functional annotation and Gene Ontology Enrichment analysis were done through VectorBase, using the LOC number of all DEGs (bulk RNA-seq) or marker genes (scRNA-seq) as input, and the following parameters: (1) Organism: "*Ixodes scapularis* PalLabHiFi"; (2) Ontology: "Molecular Function" or "Biological Process"; (3) adjusted p-value cutoff (Benjamini) $< 0.05$[67]. Additionally, the putative function of the top 50 marker genes in each single cell cluster were manually assigned based on 10 categories ("ncRNA/Pseudogene", "detoxification", "secreted protein/extracellular matrix", "metabolism", "immunity", "hormone related", "cell proliferation/differentiation", "protein synthesis", "actin polymerization/cell rearrangement", "unknown"). First, the protein coding sequence of each gene was retrieved from the NCBI website. Next, homologues from each gene were searched in the *D. melanogaster* genome using the BLAST tool from Flybase[68]. If a match was found with an E value $< 10^{-20}$, then the genes were assumed to have some (E value was between $< 10^{-20}$ and $10^{-39}$) or complete (E value $< 10^{-40}$) homology. The function of the *I. scapularis* gene was interpreted based on the description for *D. melanogaster*. Comparison to gene expression profiles in specific *D.*

*melanogaster* hemocyte subtypes was assessed using the DRSC/TRiP Functional Genomics Resources dataset[30]. Alternatively, if a high match to known *D. melanogaster* genes was not found, then the complete coding sequence of the *I. scapularis* gene was used to search for protein domains using InterPro and the functional information retrieved from Pubmed or UniProt[69,70]. Otherwise, the gene function was assigned as "unknown".

### RNA-fluorescence in situ hybridization (FISH)

The expression of specific genes in hemocytes was determined by RNA-FISH using the RNAScope-Multiplex Fluorescent Reagent Kit v2 Assay (Advanced Cell Diagnostics), following the manufacturer's instructions with slight modifications. Briefly, hemocytes were pooled from five unfed ticks in microtubes containing 100 μL of L15C300. Then, cells were spun onto microscope slides using a Cytospin 2 and fixed with 4% paraformaldehyde (PFA, Millipore-Sigma) for 20 min. After three washing steps with 1X PBS, two drops of hydrogen peroxide were applied to the samples for 10 min and washed three times with 1X PBS. Subsequently, two drops of Protease III were added to the cells and the samples incubated in the HybEZ™ Oven (Advanced Cell Diagnostics) for 30 min at 40 °C. Samples were washed three times with 1X PBS before the addition of two drops of the following RNA-Scope probes (Advanced Cell Diagnostics): *hemocytin* RNA probe (regions 2778-3632 of XM_042287086.1) conjugated with C1, *astakine* probe (regions 47-800 of XM_040222953.1) conjugated with C2, *actin5c* probe (regions 2-1766 of XM_029977298.4) conjugated with C1 as a positive control and *gfp* probe (regions 12-686 of AF275953.1) conjugated with C1 or C2 as negative controls. Slides were incubated in a HybEZ™ Oven for two hours at 40 °C. Samples were then washed three times with 1X PBS and stored overnight in 5X SSC buffer (Millipore Sigma). Next, slides were washed using 1X PBS and incubated with the AMP reagents as instructed by the manufacturer. To develop the probe signals, samples were washed three times with 1X PBS and three drops of the corresponding RNAscope Multiplex FLv2 HRP was added for 15 min at 40 °C. Then, slides were washed again with 1X PBS and incubated with ~50 μL of 1:1500 dilution of Opal™ 520 and 570 dyes (Akoya Biosciences), for C1 and C2 respectively, before the addition of two drops of the RNAscope Multiplex FLv2 HRP blocker for 15 min at 40 °C. Slides were counterstained with two drops of DAPI for 30 s and mounted using two drops of ProLong™ Gold Antifade Mountant (Thermo Scientific). Slides were allowed to dry overnight before examined under a Nikon W-1 spinning disk confocal microscope. For imaging, the following laser channels were used: 405 nm (DAPI), 488 nm (GFP, C1) and 561 nm (RFP, C2).

### RNA interference

For in vitro experiments, small interfering RNAs (siRNAs) and scrambled controls (scRNAs) designed for *astakine* and *hemocytin* were synthesized by Millipore Sigma with UU overhangs. Tick cells were seeded at $3 \times 10^5$ cells/well (24-well plate) for RNA extraction or the phagocytosis assay and $1 \times 10^6$ cells/well (6-well plate) for protein extraction. siRNAs were transfected into tick cells using Lipofectamine 3000 (Thermo Scientific) at 1 μg per mL. After 7 days, cells were either harvested or infected with *A. phagocytophilum* for 48 h. For protein extraction, cells were washed with 1X PBS, resuspended in a solution of Radio-immunoprecipitation assay (RIPA) buffer (Merck Millipore) containing protease inhibitor cocktail (Roche) and stored at −80 °C. For RNA extraction, cells were harvested in TRIzol and stored at −80 °C.

For in vivo experiments, siRNAs and scRNAs designed for *astakine* and *hemocytin* were synthesized using the Silencer™ siRNA Construction Kit (Thermo Scientific) with the primers listed in Supplementary Data 12, following the manufacturer's instruction. Tick microinjections were performed using 40–60 ng of siRNA or scRNA per tick. Ticks were allowed to recover overnight before being placed on uninfected or *A.*

*phagocytophilum*-infected mice. After placement, the percentage of ticks attached to mice was calculated. Fully fed ticks were collected 4 days post-placement, weighed, and either maintained in a humidified chamber for molting experiments or frozen at −80 °C in 200 μL TRIzol for RNA extraction.

## Phagocytosis assay

IDE12 were transfected with siRNA or scRNA designed for *hemocytin*, as previously described in ref. 6. Briefly, 7 days post-transfection, cells were re-plated at a density of $4 \times 10^5$ cells/well in a 24-well plate and left to adhere overnight. The following day, media was replaced with 500 μL of L15C300 containing a 1:10,000 dilution of FluoSpheres™ Carboxylate 1.0 μm beads (yellow-green, Invitrogen) for 24 h. Cells were then washed five times with 1X PBS and fixed in 200 mL of 4% PFA for 20 min. Fluorescence and bright field microscopy images were acquired with an EVOS FL Digital Inverted Microscope (Advanced Microscopy Group) and merged using ImageJ software. Negative and positive cells for fluorescent microspheres and the average number of phagocytosed beads per cell were calculated from the number of green puncta detected in the total cells counted in each field (>100 cells). Two independent experiments were performed, with fields taken from three different wells per replicate.

## Recombinant astakine

*I. scapularis* recombinant Astakine (rAstk) was produced by Gene-Script, based on the coding sequence of the XM_040222953.1 *astakine* transcript (LOC8032444). For in vitro cell proliferation assays, $1 \times 10^5$ IDE12 cells/well were seeded in L15C300 complete medium (24-well plate). The following day, cells were treated with 0.05 μg/mL of rAstk for a total of 5 days. Cells were collected in a 1.5 mL microtube and 10 μL of the suspension was used for live cell counting using the trypan blue exclusion assay with a TC20™ automated cell counter (Bio-Rad). BSA (0.05 μg/mL) was used as negative control. For in vivo experiments, nymphs were microinjected with 50 nL of PBS containing 0.05 ng, 0.5 ng or 5 ng of rAstk[26–28]. Afterwards, ticks were allowed to rest for 24 h before hemolymph was collected, as described above. Microinjection of BSA (5 ng) was used as a negative control.

## Clodronate depletion of phagocytic hemocytes

To determine the effect of phagocytic hemocytes in *I. scapularis* nymphs, we used clodronate liposomes (CLD) to deplete phagocytic cells[35–37]. Ticks were immobilized on a glass slide using double-sided tape and microinjected (Nanojet III microinjector; Drummond Scientific) through the anal pore with 69 nL of either CLD (Standard macrophage depletion kit, Encapsula NanoSciences LLC) or control liposomes (LP) at a 1:5 dilution in 1X PBS. Ticks were allowed to recover overnight before being placed on uninfected or *A. phagocytophilum*-infected mice. After placement, the percentage of ticks attached to mice was calculated. Fully fed ticks were collected 4 days post-placement, weighed, and either maintained in a humidified chamber for molting experiments or frozen at −80 °C in 200 μL TRIzol for RNA extraction.

## RNA extraction and quantitative real-time PCR (qPCR)

Total RNA from cell lines or engorged ticks was isolated from samples preserved in TRIzol using the PureLink RNA Mini kit (Ambion). Total RNA from tick hemocytes was purified following the manufacturer's instruction for TRIzol RNA extraction with the addition of 1 μg of glycogen (RNA grade, Thermo Scientific). For cDNA synthesis, 400–600 ng of RNA was used in the Verso cDNA Synthesis Kit (Thermo Scientific). To measure gene expression by RT-qPCR, 2X Universal SYBR Green Fast qPCR Mix (ABclonal) was used with the addition of 1 μL of cDNA per well and quantified using a CFX96 Touch Real-Time PCR Detection System (Bio-rad). All genes of interest were amplified at

54 °C, except for amplification of *A. phagocytophilum* 16 S, which was run at 46 °C. No-template controls were included to verify the absence of primer-dimers formation and/or contamination. Reactions on each sample and controls were run in duplicate. Gene expression was determined by relative quantification normalized to tick *actin*, using the primers described in Supplementary Data 12.

## Western blotting

To extract proteins, wells were washed with cold 1X PBS, and resuspended in 1X RIPA buffer (Merck Millipore) containing protease inhibitor cocktail (Roche). Protein extracts were stored at −80 °C and their concentration was estimated using Bicinchoninic acid (BCA) assay (Thermo Scientific). For sodium dodecyl sulfate polyacrylamide gel electrophoresis (SDS-PAGE), equal amounts of protein were boiled in 6X Laemmli sample buffer (Alfa Aesar) containing 5% β-mercaptoethanol. The proteins were then transferred onto PVDF membranes (Biorad) and blocked for 1 h using a solution of 5% skim milk prepared in 1X PBS and 0.1% Tween® 20 Detergent (PBS-T). Primary antibodies were incubated overnight at 4 °C, followed by four washes in PBS-T and incubation with secondary antibodies for at least 1 h at room temperature with gentle agitation. The blots were washed four times in PBST, incubated with enhanced chemiluminescence (ECL) substrate solution for 1 min (Millipore), and then imaged. A complete list of antibodies used can be found in Supplementary Data 13.

## CRISPR activation (CRISPRa)

The SP-dCas9-VPR plasmid was a gift from George Church (Addgene plasmid, #63798). For expression of sgRNAs, the sgRNA for the *hemocytin* promoter or a control sequence (random sequence that does not match with any region in the *I. scapularis* genome) was cloned between the *BbsI* sites of a derivative of the pLib8 vector (pLibT-B.1_ISCW_025) under the control of an *I. scapularis* U6 promoter (ISCW025025) (Supplementary Fig. 20). ISE6 cells ($7 \times 10^7$ cells) were nucleofected with 15 μg of SP-dCas9-VPR plasmid using the buffer SF, pulse code EN150 (Lonza Biosciences) and subsequently incubated for 3 days. Following incubation, the cells were selected with neomycin (250 μg/mL) for 2 weeks. After antibiotic selection, the cells were nucleofected with 1 μg of the pLibTB.1_ISCW_025 vector containing the experimental or scrambled control sgRNA and were harvested 3 days later. Cells were collected in TRIzol for RNA extraction, as previously described.

## Statistics

Statistical significance for quantitative variables was tested with an unpaired two-tailed t-test with Welch's correction, two-tailed Mann–Whitney U test or one-way analysis of variance (ANOVA) followed by Tukey's multiple comparison test when appropriate. Gaussian distribution was assessed using the D'Agostino-Pearson omnibus K2 test, and homogeneity of variance was determined by F-test when comparing two conditions or the Brown-Forsythe test when comparing three conditions. Statistical significance for categorical variables was assessed using Fisher's exact test. GraphPad PRISM® (version 9.1.2) was used for all statistical analyses. Outliers were detected by a Grubbs' Test, using the GraphPad Quick Cals Outlier Calculator. $p$ values of <0.05 were considered statistically significant.

## Reporting summary

Further information on research design is available in the Nature Portfolio Reporting Summary linked to this article.

# Data availability

All primers, reagents, resources, and software used in this study, together with their manufacturer's information and catalog numbers are available in Supplementary Data 12 and 13. Source data are

provided with this article. All sequencing reads for the bulk RNA sequences were deposited into the NCBI Sequence Read Archive under the BioProject accession PRJNA906572. All scRNA sequences were deposited into the NCBI Sequence Read Archive under the BioProject accession PRJNA905678. Further information and request for resources and reagents should be directed to and will be honored by the corresponding author: Joao HF Pedra (jpedra@som.umaryland.edu). Source data are provided with this paper.

## Code availability

The codes for the scRNA sequence analysis are available through the GitHub websites: https://github.com/Haikelnb/scRNASeq_Analysis and https://github.com/HLaukaitisJ/PedraLab_hemocyte_scRNAseq[71].

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

## Acknowledgements

This work was funded by the National Institutes of Health (NIH) grants T32AI162579 (HJL-Y), F31AI152215 (A.J.O.), F31AI167471 (L.R.B.), R01AI134696 (J.H.F.P.), R01AI116523 (J.H.F.P.), P01AI138949 (J.H.F.P., U.P.) and R21 AI168592 (N.P., S.E.M., J.H.F.P.). Funding was also provided by the Fairbairn Family Lyme Research Initiative (N.P., S.E.M., and J.H.F.P.) and the Howard Hughes Medical Institute (N.P.). The authors acknowledge members of the Pedra laboratory for providing insightful discussions and colleagues for manuscript feedback. We thank Jon Skare (Texas A&M University Health Science Center) for providing the *B. burgdorferi* B31 strain, clone MSK5; Ulrike G. Munderloh (University of Minnesota) for supplying ISE6 and IDE12 tick cells; Joseph Mauban (University of Maryland School of Medicine) for aiding in the microscopy analysis; Jonathan Oliver (University of Minnesota) for providing available *I. scapularis* ticks; Dana Shaw (Washington State University) and Adela Oliva Chavez (Texas A&M University) for information regarding cultures of *B. burgdorferi* and *A. phagocytophilum*; the University of Maryland, Baltimore Confocal Microscopy Core and the Maryland Genomics Core at the Institute for Genome Sciences for the services provided in microscopy and next generation sequencing, respectively. Some images were created with BioRender.com. The content is solely the responsibility of the authors and does not represent the official views of the NIH, the Department of Health and Human Services or the United States government.

## Author contributions

A.R. and J.H.F.P. designed the study. A.R., H.J.L.-Y., H.N.B., N.S., S.S., M.B., C.R.F., A.J.O., M.T.M. and L.M. performed the experiments. A.R., H.J.L.-Y., and H.N.B. performed computational analysis. M.B., N.S., E.M., and B.X. established the CRISPRa in the tick system. A.R., H.J.L.-Y., A.J.O. and J.H.F.P. wrote the manuscript. L.R.B., L.M.V., V.S.R. and F.E.C.P. aided with experimentation. L.R.B. created some schematics. All authors analyzed the data, provided intellectual input into the study, and contributed to editing of the manuscript. C.S., S.E.M., U.P., N.P. and D.S. supervised experiments and provided instruments and reagents. J.H.F.P. supervised the study.

## Competing interests

The authors declare no competing interests.
