## [Peer Review File · Nature Communications]

Tick hemocytes have a pleiotropic role in microbial infection and arthropod fitnessREVIEWER COMMENTS

Reviewer #1 (Remarks to the Author):

Ticks are important vectors transmitting many tick-borne diseases, which are posing severe threat to public health. The hemocytes are major immune cells in arthropods including ticks, they defense ticks from various pathogens, but their types, functions, character are not well understood yet. This study investigated the tick hemocyte using single cell RNA-Seq and other multiple cutting-edge techniques including CRISPR/Cas9a, identified the 6 and 12 clusters from hemocytes of the unfed tick and fed ticks, found that blood feeding can change cellular composition and induced new cell clusters, *A. phagocytophilum* and *B. burgdorferi* affect the molecular signatures of hemocytes in fed *I. scapularis* nymphs, the study further discovered that Hemocytin affects the JNK pathway of *I. scapularis*, Astakine induces hemocyte proliferation and differentiation in *I. scapularis*. The study presented a cell atlas of tick hemocytes, enriched the knowledge of arthropod hemocyte and immune system, provided an excellent base for studying tick immune system and tick-pathogen interaction in future. I strongly suggest that Nature Communication should publish the novel study.

There are a few minor comments/questions below need to be replied,

Line 152, Based on similarities in marker gene profiles, two clusters in the unfed and two clusters in the engorged datasets were merged (Datasets S3-4). Which clusters are merged? It is hard to see it.

In Fig 7C, E, F, all the values (cycles) of tick weight looks integrators (1, 2, 3, 4, 5, 6 mg), which should be not happen, could you explain this?

The scRNA-seq of hemocyte of fly and mosquitoes including *Anopheles* and *Aedes*, especially cell clusters of the fly hemocyte are well annotated with traditional cell type classification, are you able to annotate some clusters with traditional cell type classification? It will be great if you can compare and discuss scRNA studies of ticks, fly and mosquitoes, show the conservation and difference among these arthropod vectors.

Reviewer #2 (Remarks to the Author):

I have thoroughly reviewed this manuscript focusing on the immune cells or hemocytes of the *Ixodes scapularis* tick. The study is indeed a significant contribution to the field of arthropod immunology, particularly in the context of non-model organisms. The use of advanced techniques like bulk and single-cell RNA sequencing, CRISPRa, and RNA-FISH adds substantial depth to authors' findings. This work stands out for its detailed analysis of the molecular and cellular mechanisms underpinning tick immunity and its implications for understanding the broader biological processes in arthropods. I recommend its publication with revisions.

I have three major comments and a few minor comments.

Specific Comments:

1. Abstract Clarity: The abstract is dense with information. A more streamlined approach, focusing on key findings and their broader implications, might increase its impact.
2. Conclusions: The conclusions drawn are well-supported by your data. However, integrating a discussion on the potential applications of these findings in arthropod biology or disease control could add value.
3. Future Directions: While you hint at future research avenues, explicitly stating potential future experiments or hypotheses based on your findings could be beneficial.
4. I found a few issues with language. I would suggest revising the wording carefully. Here are a few examples.

Line 45. The phrase "which impacts" could be revised to "impacting" for smoother flow.

Line 61. "janus kinase/signal transducer and activator of transcription" should be capitalized as "Janus Kinase/Signal Transducer and Activator of Transcription (JAK/STAT)."

Line 74. Correction: Change "remain" to "remains" to agree with the singular subject

"understanding".

Line 90. Consider using "affects" instead of "impacted" for clearer meaning.

Line 139. the verb "is" should be "are" to agree with the plural subject "transcriptional changes."

Line 200. Briefly explaining how pseudotime analysis aids in predicting hemocyte ontogeny could enhance understanding.

Line 226. "Collectively, both *A. phagocytophilum* ... *I. scapularis* nymphs", this sentence could be clearer. It might be better to specify how these agents affect the molecular signatures, or at least indicate that the details are discussed in the subsequent text.

Line 231. For smoother flow, consider rephrasing to: "Therefore, our focus shifted to two marker genes from the Immune 1 hemocyte cluster: hemocytin and astakine (Fig. 4A, Datasets S5-6)."

Line 279. "We optimized reagents ... RNA polymerase III promoter (Fig. 5C)." This sentence is quite long and complex. Breaking it into two sentences might improve readability.

Line 287. It could be more clear to state "elevated levels of both jun (the transcription factor for the JNK pathway) and jnk expression."

Line 309. "Corroborate" is a more precise term than "support" in this context.

Line 322. "We found that ... hemocytes in nymphs (Fig. 7A)." This sentence is correct, but it might be more informative if you specify how significantly the number was reduced or provide a brief insight into the implications.

Line 331. Add "the" before "Immune"

Line 358. The phrase "it is conceivable" could be replaced with "we hypothesize" or "our data suggest" for a more assertive tone.

Line 367. The phrase "non-lethal relationship" is a bit vague. Consider specifying what this relationship entails for clarity.

In conclusion, this manuscript represents a significant step forward in our understanding of tick immunology. With a few suggested improvements, it has the potential to make a substantial impact in the field. I look forward to seeing the revised manuscript.

Reviewer #3 (Remarks to the Author):

Roladelli and colleagues used (and created) many sophisticated technologies to evaluate the role of tick hemocytes on anaplasma and borrelia colonization, in addition to tick fitness. While there are certainly some notable achievements, this reviewer's enthusiasm is dampened by the overall clarity and robustness of some analyses. It seems as though this is a bit of a data dump without clear objectives and a way to showcase a couple new methods.

At the risk of being blunt, this reviewer would be more enthusiastic if the story and material were presented differently and it didn't feel like the message was '... we perturbed the system, things changed, not sure what they mean, but a couple of genes are probably important'

Major Issues

-There is a big missed opportunity to highlight some important morphological aspects of hemocytes subpopulations in Fig 1B. General features in terms of size are made, but no graphical representation of the data or stats. A single cell is not representative of population level statements. It would also be comforting to understand the QC associated with the preparations, other cell types present, etc (and see below).

-Line 157-159: My interpretation of the data exclusion is that there is contamination. What's the frequency? How much? How is the reader supposed to interpret the bulk RNA seq data with this in mind. This is concerning and certainly needs to be addressed.

-Related to above, it seems a bit conspicuous that partially fed nymphs are included in Fig 1C, but excluded in D and E. There are fascinating trends and this is clearly a true intermediate phenotype as it relates to the data shown in 1C.

-Fig. 3D doesn't make sense.

-Fig. 4B is not interpretable. No clear controls, a few cells shown, but statements are made about the percent of cells expressing both hmc and astk. This experimental approach, data presented, and analysis of these data requires considerable revisions.

Minor Issues

- All of the Figure legends need more information and should be re-organized for clarity purposes. For instance, lines 766-768 don't make sense.
- Line 68-70: This is misleading. The focus is clearly on Ixodes/hard ticks, not soft ticks (which have an expanded lifespan).
- Line 101: Presumptuous
- The way Fig. 3A is organized is difficult to follow and could be presented very differently to improve clarity (see Fig 5D and E for something that would be more digestible).
- The use of the term 'sc-hmc' as it relates to scrambled RNA is misleading, because it seems as though it is specifically scrambled for the hmc locus. Instead, as far as I can tell in the methods, it's just a random RNA.
- The references are thin and the scientific names aren't italicized.
- The reagents table has typos and includes the use of NA inappropriately in some places.

REVIEWER COMMENTS

Reviewer #1 (Remarks to the Author)

Ticks are important vectors transmitting many tick-borne diseases, which are posing severe threat to public health. The hemocytes are major immune cells in arthropods including ticks, they defend ticks from various pathogens, but their types, functions, character are not well understood yet. This study investigated the tick hemocyte using single cell RNA-Seq and other multiple cutting-edge techniques including CRISPR/Cas9a, identified the 6 and 12 clusters from hemocytes of the unfed tick and fed ticks, found that blood feeding can change cellular composition and induced new cell clusters, *A. phagocytophilum* and *B. burgdorferi* affect the molecular signatures of hemocytes in fed *I. scapularis* nymphs, the study further discovered that hemocytin affects the JNK pathway of *I. scapularis*, astakine induces hemocyte proliferation and differentiation in *I. scapularis*. The study presented a cell atlas of tick hemocytes, enriched the knowledge of arthropod hemocyte and immune system, provided an excellent base for studying tick immune system and tick-pathogen interaction in future. I strongly suggest that *Nature Communications* should publish the novel study. There are a few minor comments/questions below need to be replied.

1. Line 152, Based on similarities in marker gene profiles, two clusters in the unfed and two clusters in the engorged datasets were merged (Datasets S3-4). Which clusters are merged? It is hard to see it.

The clusters that were merged from the initial analysis are Cluster 1 and Cluster 4 of the flat tick hemocytes dataset, and Cluster 2 and Cluster 10 of the engorged tick hemocytes dataset. The similarity in marker genes between those merged clusters is shown in Supplementary Data 3-4. Following the reviewer's comment and to make the merged clusters easily identifiable, we have highlighted each with an asterisk in Supplementary Fig. 6 and added the corresponding description in the revised figure legend.

In Fig 7C, E, F, all the values (cycles) of tick weight looks integrators (1, 2, 3, 4, 5, 6 mg), which should be not happen, could you explain this?

The recorded tick weights are restricted to the resolution of our weight scale, which is 0.001g (or 1mg). Therefore, all recorded weight values are at the integer level.

The scRNA-seq of hemocyte of fly and mosquitoes including *Anopheles* and *Aedes*, especially cell clusters of the fly hemocyte are well annotated with traditional cell type classification, are you able to annotate some clusters with traditional cell type classification? It will be great if you can compare and discuss scRNA studies of ticks, fly and mosquitoes, show the conservation and difference among these arthropod vectors.

In the revised discussion, we have now speculated which hemocyte clusters correspond to the most probable morphotypes (lines 398-410). We added two additional paragraphs comparing scRNA-seq studies of ticks, flies and mosquitoes, and elaborated on the conservation and differences among them (lines 412-439). We also discussed how the technologies developed in our manuscript could be used to improve hemocyte annotation within the community (lines 492-504).

Reviewer #2 (Remarks to the Author)

I have thoroughly reviewed this manuscript focusing on the immune cells or hemocytes of the *Ixodes scapularis* tick. The study is indeed a significant contribution to the field of arthropod immunology, particularly in the context of non-model organisms. The use of advanced techniques like bulk and single-cell RNA sequencing, CRISPRa, and RNA-FISH adds substantial depth to authors' findings. This work stands out for its detailed analysis of the molecular and cellular mechanisms underpinning tick immunity and its implications for understanding the broader biological processes in arthropods. I recommend its publication with revisions. I have three major comments and a few minor comments.

1. Abstract Clarity: The abstract is dense with information. A more streamlined approach, focusing on key findings and their broader implications, might increase its impact.

We have modified its content in the revised version of the manuscript (lines 39-51).

2. Conclusions: The conclusions drawn are well-supported by your data. However, integrating a discussion on the potential applications of these findings in arthropod biology or disease control could add value.

We expanded the last paragraph in the discussion section of the revised manuscript (lines 493-505). We specifically address the potential applications of our findings and developed technologies in ticks. We believe that these applications will promote hypothesis-driven research aligned and be beneficial for the wider arthropod immunology community.

3. Future Directions: While you hint at future research avenues, explicitly stating potential future experiments or hypotheses based on your findings could be beneficial.

We have outlined new questions arising from our findings and future research directions. The revised discussion now highlights anticipated experiments and hypotheses that can be derived from our current findings. We emphasize the potential functions of different hemocyte clusters and the pleiotropic role of phagocytic hemocytes, as exemplified by the function of *hemocytin* and *astakine* in various aspects of tick physiology (lines 458-490).

4. I found a few issues with language. I would suggest revising the wording carefully. Here are a few examples.

We have thoroughly revised the manuscript for typos and language issues, including the grammatical errors.

Line 45. The phrase "which impacts" could be revised to "impacting" for smoother flow.

The modification was made accordingly (line 47).

Line 61. "janus kinase/signal transducer and activator of transcription" should be capitalized as "Janus Kinase/Signal Transducer and Activator of Transcription (JAK/STAT)."

The modification was made accordingly (lines 62-64).

Line 74. Correction: Change "remain" to "remains" to agree with the singular subject "understanding".

This word was modified (lines 77).

Line 90. Consider using "affects" instead of "impacted" for clearer meaning.

The suggested replacement was made (lines 92).

Line 139. the verb "is" should be "are" to agree with the plural subject "transcriptional changes."

The suggested replacement was made (lines 147).

Line 200. Briefly explaining how pseudotime analysis aids in predicting hemocyte ontogeny could enhance understanding.

We have added additional details of pseudotime prediction of cell lineages and have referenced the relevant article (lines 212-214). Pseudotime is a computational method in which cells can be ordered across lineages based on transcriptional profiles. For instance, single-cell datasets denote heterogeneous populations that often lack clear distinctions between cellular states. Instead, cells represent points across a continuum with gradual transcriptional changes, rather than discrete steps of differentiation. Therefore, pseudotime interference identifies lineages (ordered sets of cell clusters) where each share a starting cluster and a unique terminal cluster. Then, pseudotime is calculated based on each cell's transcriptional progression toward the terminal state.¹

Line 226. "Collectively, both *A. phagocytophilum* ... *I. scapularis* nymphs", this sentence could be clearer. It might be better to specify how these agents affect the molecular signatures, or at least indicate that the details are discussed in the subsequent text.

We have expanded on the number of genes altered specifically by *Anaplasma* or *Borrelia* for each of the Immune, Metabolism or Proliferative hemocyte groups (lines 235-246, revised manuscript). The list of individual genes modified by *Anaplasma* or *Borrelia* are listed in Supplementary Data 10. Additionally, we have changed Fig. 3d to represent these results in a clearer way (see also the response to comment #4 from Reviewer 3).

Line 231. For smoother flow, consider rephrasing to: "Therefore, our focus shifted to two marker genes from the Immune 1 hemocyte cluster: hemocytin and astakine (Fig. 4A, Datasets S5-6)."

This sentence was rephrased, as suggested (lines 250-252).

Line 279. "We optimized reagents ... RNA polymerase III promoter (Fig. 5C)." This sentence is quite long and complex. Breaking it into two sentences might improve readability.

We modified this sentence for clarity as suggested (lines 359-364).

Line 287. It could be more clear to state "elevated levels of both jun (the transcription factor for the JNK pathway) and jnk expression."

The suggested replacement was made (lines 367-369).

Line 309. "Corroborate" is a more precise term than "support" in this context.

This word was modified (line 320).

Line 322. "We found that ... hemocytes in nymphs (Fig. 7A)." This sentence is correct, but it might be more informative if you specify how significantly the number was reduced or provide a brief insight into the implications.

We have now added the percent reduction in total hemocyte numbers detected after CLD treatment (lines 271).

Line 331. Add "the" before "Immune"

This word was added (line 276).

Line 358. The phrase "it is conceivable" could be replaced with "we hypothesize" or "our data suggest" for a more assertive tone.

The suggested replacement was made (line 400).

Line 367. The phrase "non-lethal relationship" is a bit vague. Consider specifying what this relationship entails for clarity.

This sentence was rephrased for clarity (lines 441-442).

In conclusion, this manuscript represents a significant step forward in our understanding of tick immunology. With a few suggested improvements, it has the potential to make a substantial impact in the field. I look forward to seeing the revised manuscript.

We greatly appreciate the reviewer's constructive feedback and the recognition of our work's significance.

Reviewer #3 (Remarks to the Author)

Rolandelli and colleagues used (and created) many sophisticated technologies to evaluate the role of tick hemocytes on *Anaplasma* and *Borrelia* colonization, in addition to tick fitness. While there are certainly some notable achievements, this reviewer's enthusiasm is dampened by the overall clarity and robustness of some analyses. It seems as though this is a bit of a data dump without clear objectives and a way to showcase a couple new methods. At the risk of being blunt, this reviewer would be more enthusiastic if the story and material were presented differently and it didn't feel like the message was '... we perturbed the system, things changed, not sure what they mean, but a couple of genes are probably important'

We appreciate the reviewer's candid feedback on our manuscript. We understand the reviewer's concerns regarding clarity and robustness, and we apologize for any ambiguity in presenting our story. We addressed these issues in the revised version, with a focus on providing a more structured and cohesive narrative by presenting the story in a different way, encompassing the characterization of tick hemocytes from a global scale to a molecular level. As a result, Figures 4 to 7 of the original version were reshuffled and more results were included (Fig. 1d, 4g, 4i and Supplementary Fig. 1, 3, 10). The story narrative now explores hemocytes as whole (Figure 1), then moves to the characterization of hemocyte subpopulations (Figures 2-3), including functional analysis of phagocytic hemocytes (Figure 4), and it finalizes with the molecular characterization of the hemocyte markers *hemocytin* and *astakine* (Figures 5-6). We have refined the manuscript to better convey the significance of our findings by clarifying objectives and highlighting the novelty of our methods. Importantly, we have streamlined the content of the abstract and expanded on the discussion regarding potential applications of our findings and future directions. We believe that we are now presenting a more cohesive and impactful story in the revised manuscript.

Major Issues:

-There is a big missed opportunity to highlight some important morphological aspects of hemocytes subpopulations in Fig 1B. General features in terms of size are made, but no graphical representation of the data or stats. A single cell is not representative of population level statements. It would also be

comforting to understand the QC associated with the preparations, other cell types present, etc (and see below).

In the revised manuscript, we have included a new figure (Supplementary Fig. 1), which contains several images of individual prohemocytes, plasmatocytes and granulocytes (panels a-c). Additionally, we measured the diameter of different cells ($n=33-36$) belonging to each morphotype and visually displayed the summary in panel d. Thus, this new figure reflects a more defined description of the morphological aspects of hemocyte subpopulations discussed in this manuscript. The differences in cell size are clearly depicted, and a more detailed description of morphological differences within hemocytes was added to the text (lines 107-117).

The reviewer brings attention to an important topic of cellular contamination. We occasionally observed some cells not indicative of hemocytes (*i.e.*, clusters of cells or tissue-like structures), which were not included in the hemocyte data presented. Although these debris could be considered “contaminants”, they are also commonly present in most hemolymph collection protocols described for arthropods.²⁻⁶ These cells represented a minor fraction of the sample (see below) and were not considered in downstream analyses. However, we now refer to our samples as “hemocyte-enriched” in the revised version of the manuscript.

-Line 157-159: My interpretation of the data exclusion is that there is contamination. What's the frequency? How much? How is the reader supposed to interpret the bulk RNA seq data with this in mind. This is concerning and certainly needs to be addressed.

The frequency of non-hemocyte cells represents a minor fraction of the sample. We have now quantified the number of cells and the percentage of all the clusters obtained in the scRNA-seq analysis, including non-hemocyte cells. This information is available as a new supplementary dataset (Supplementary Data 5). Additionally, we have included the percentage of each cluster in Fig. 2a-b. As shown, the percentage of non-hemocyte cells in hemocyte-enriched samples from flat ticks is 0.65%, while in hemocyte-enriched samples from engorged ticks (infected and uninfected) is 1.56%. We were able to exclude these cells from the scRNA-seq dataset using computational approaches, but we cannot exclude them from the bulk RNA-seq analysis. Nevertheless, as the frequency of non-hemocyte cells is minor, we believe that they did not skew the results of the bulk RNAseq. Besides, the functional enrichment obtained after analyzing the bulk RNAseq dataset in hemocyte-enriched samples from engorged ticks (over-representation of genes related to (1) immunity; (2) protein catabolism; (3) lipid, nucleotide and carbohydrate anabolism; (4) cell proliferation and (5) development) are corroborated by the clusters and the marker genes present in the scRNA-seq analysis.

-Related to above, it seems a bit conspicuous that partially fed nymphs are included in Fig 1C, but excluded in D and E. There are fascinating trends and this is clearly a true intermediate phenotype as it relates to the data shown in 1C.

We concur with the reviewer's observation that hemocytes from partially fed ticks exhibit an intermediate phenotype in terms of both numbers and morphotype proportions compared to unfed and engorged ticks. Consequently, as hemocyte-enriched samples from unfed and engorged ticks represent the distinct phenotypes, we focused on these samples for bulk RNAseq to capture the most pronounced biological signatures that change during feeding. In response to the reviewer's comment, we have now included the results of partially fed ticks in Fig. 1d and Supplementary Fig. 3. We have modified the results section accordingly (lines 136-143). The expression of *croquemort* (*crq*) in hemocyte-enriched samples from partially fed ticks compared to unfed ticks was previously published by our group.⁷ To avoid any duplication in publications, we excluded these results from the revised version of the manuscript. However, the reviewer can see the graph compiling *crq* expression in the samples below.

Expression profile of *croquemort* in hemocytes-enriched samples from *I. scapularis* nymphs. The expression level of *croquemort* (*crq*) was evaluated in hemocyte-enriched samples from unfed (ivory), partially fed (light blue) and engorged (dark blue) ticks by RT-qPCR ($n=5-10$; samples comprising 40-80 pooled ticks). Results represent mean \pm SD. A minimum of 5 independent experiments were performed. Statistical significance was evaluated by Brown-Forsythe ANOVA test. * $p<0.05$; **** $p<0.0001$.

-Fig. 3D doesn't make sense.

We apologize for the lack of clarity. We have now modified this figure in the revised manuscript to show two Venn diagrams informing readers of the numbers of DEGs per cluster grouping, one for *Anaplasma* infection and one for *Borrelia* infection (results described in lines 235-244). To complement this figure, the complete list of genes differentially expressed are shown in the Supplementary Data 10. Overall, we believe that this figure highlights that *Anaplasma* and *Borrelia* infection differentially affect gene expression in hemocytes, some of which are modulated in specific clusters while others are shared between groups. We hope that the reviewer finds the revised version of the figure appropriate.

-Fig. 4B is not interpretable. No clear controls, a few cells shown, but statements are made about the percent of cells expressing both *hmc* and *astk*. This experimental approach, data presented, and analysis of these data requires considerable revisions.

In the revised manuscript, we modified a previous statement and included a new figure representing control images for RNA FISH (Supplementary Fig. 10). We used a Channel 1 (C1) probe recognizing the *I. scapularis actin5c* gene to demonstrate the detection of positive dots through RNA FISH (positive control); and two probes (C1 and C2) targeting the *gfp* gene (absent from the *I. scapularis* genome) to illustrate the absence of non-specific signals from cells (negative controls). The primary goal of Figure 4b is to illustrate that *hemocytin* and *astakine* can serve as marker genes of hemocyte subpopulations. This is evident by the fact that some hemocytes express these markers, as shown by RNA FISH in the image, thereby validating the results of the scRNAseq analysis. The revised version of our manuscript also discusses potential applications of our findings (lines 492-504).

Minor Issues

-All of the Figure legends need more information and should be re-organized for clarity purposes. For instance, lines 766-768 don't make sense.

We have revised all figure legends and provided additional details for clarity.

-Line 68-70: This is misleading. The focus is clearly on *Ixodes*/hard ticks, not soft ticks (which have an expanded lifespan).

We have modified the sentence to emphasize our focus is on hard ticks (lines 70-72).

-Line 101: Presumptuous

We toned down the language (lines 102-104).

-The way Fig. 3A is organized is difficult to follow and could be presented very differently to improve clarity (see Fig 5D and E for something that would be more digestible).

We have presented Fig. 3a of the revised manuscript similar to Fig. 5d-e of the initial submission.

-The use of the term 'sc-hmc' as it relates to scrambled RNA is misleading because it seems as though it is specifically scrambled for the *hmc* locus. Instead, as far as I can tell in the methods, it's just a random RNA.

We used the term *sc-hmc* because it is indeed a scrambled RNA sequence of the siRNA designed specifically for the *hmc* locus. The same for *sc-astk*. However, we reexamined the sequences shown in the Supplementary Table 1 and found a mistake in the scHmc_R sequence. The scHmc_R sequence in the Supplementary Table 1 of the revised version was modified appropriately. The sgRNA used as control for CRISPRa experiments is a random sequence, not specifically scrambled for the *hmc*-sgRNA. Thus, we have changed the term "scrambled-sgRNA" for "control-sgRNA" in the revised version of the manuscript (Fig. 6d and Supplementary Table 2).

-The references are thin and the scientific names aren't italicized.

We have now ensured that all scientific names are properly italicized and added additional supportive citations, without exceeding the 70 references instructed by the journal.

-The reagents table has typos and includes the use of NA inappropriately in some places.

In the revised version of our manuscript, we have thoroughly reviewed and made the appropriate corrections in the reagents table.

In closing, we are excited that the comments of the reviewers made our story much better. We thank *Nature Communications* for the manuscript evaluation.

Sincerely yours,

Joao Pedra, PhD
Professor of Microbiology and Immunology

References

- 1 Street, K. *et al.* Slingshot: cell lineage and pseudotime inference for single-cell transcriptomics. *BMC genomics* **19**, 477 (2018). <https://doi.org/10.1186/s12864-018-4772-0>
- 2 Tattikota, S. G. *et al.* A single-cell survey of *Drosophila* blood. *eLife* **9** (2020). <https://doi.org/10.7554/eLife.54818>
- 3 Leitao, A. B. *et al.* Constitutive activation of cellular immunity underlies the evolution of resistance to infection in *Drosophila*. *eLife* **9** (2020). <https://doi.org/10.7554/eLife.59095>
- 4 Kwon, H., Mohammed, M., Franzen, O., Ankarklev, J. & Smith, R. C. Single-cell analysis of mosquito hemocytes identifies signatures of immune cell subtypes and cell differentiation. *eLife* **10** (2021). <https://doi.org/10.7554/eLife.66192>
- 5 Yang, P. *et al.* Single-cell RNA sequencing analysis of shrimp immune cells identifies macrophage-like phagocytes. *eLife* **11** (2022). <https://doi.org/10.7554/eLife.80127>
- 6 Raddi, G. *et al.* Mosquito cellular immunity at single-cell resolution. *Science* **369**, 1128-1132 (2020). <https://doi.org/10.1126/science.abc0322>
- 7 O'Neal, A. J. *et al.* Croquemort elicits activation of the immune deficiency pathway in ticks. *Proceedings of the National Academy of Sciences of the United States of America* **120**, e2208673120 (2023). <https://doi.org/10.1073/pnas.2208673120>

REVIEWERS' COMMENTS

Reviewer #1 (Remarks to the Author):

The manuscript was revised according my comments, so can be accepted now.

Reviewer #2 (Remarks to the Author):

To the editor and the authors of the manuscript:

Title:[Tick hemocytes have a pleiotropic role in microbial infection and arthropod fitness]

Decision: Accept

Following my review of the revised submission, I am satisfied with the changes and improvements made by the authors in response to my initial comments and suggestions.

The authors have diligently addressed each of the concerns I raised in my previous review. Given the thoroughness of the revisions and the quality of the work presented, I am pleased to recommend that the manuscript be accepted for publication in Nature Communications. The authors have demonstrated a commendable commitment to enhancing their work based on peer feedback, and I believe the paper will be of interest to the journal's readership.

Thank you for the opportunity to review this manuscript. I appreciate the authors' efforts to improve their submission and commend them on their work. Please do not hesitate to contact me if further input is required.

Reviewer #3 (Remarks to the Author):

The authors have addressed all the concerns raised and did an excellent job.

REVIEWERS' COMMENTS

Reviewer #1 (Remarks to the Author):

The manuscript was revised according to my comments, so can be accepted now.

We appreciate the reviewer's recommendation for publication in *Nature Communications*.

Reviewer #2 (Remarks to the Author):

To the editor and the authors of the manuscript:

Title: Tick hemocytes have a pleiotropic role in microbial infection and arthropod fitness

Decision: Accept

Following my review of the revised submission, I am satisfied with the changes and improvements made by the authors in response to my initial comments and suggestions. The authors have diligently addressed each of the concerns I raised in my previous review. Given the thoroughness of the revisions and the quality of the work presented, I am pleased to recommend that the manuscript be accepted for publication in *Nature Communications*. The authors have demonstrated a commendable commitment to enhancing their work based on peer feedback, and I believe the paper will be of interest to the journal's readership. Thank you for the opportunity to review this manuscript. I appreciate the authors' efforts to improve their submission and commend them on their work. Please do not hesitate to contact me if further input is required.

We wanted to express our sincere gratitude for the reviewer's recognition of our efforts towards enhancing our work based on peer feedback and for recommending publication in *Nature Communications*.

Reviewer #3 (Remarks to the Author):

The authors have addressed all the concerns raised and did an excellent job.

We appreciate the reviewer's positive assessment of revisions made during the peer review process.